# Molecular basis of human nuclear and mitochondrial tRNA 3' processing

Arjun Bhatta[1,2,3,7], Bernhard Kuhle[1,2,7], Ryan D. Yu[1,2,3,7], Lucas Spanaus[1,2], Katja Ditter[1,2], Katherine E. Bohnsack [4] & Hauke S. Hillen [1,2,5,6] ✉

Eukaryotic transfer RNA (tRNA) precursors undergo sequential processing steps to become mature tRNAs. In humans, ELAC2 carries out 3' end processing of both nucleus-encoded (nu-tRNAs) and mitochondria-encoded (mt-tRNAs) tRNAs. ELAC2 is self-sufficient for processing of nu-tRNAs but requires TRMT10C and SDR5C1 to process most mt-tRNAs. Here we show that TRMT10C and SDR5C1 specifically facilitate processing of structurally degenerate mt-tRNAs lacking the canonical elbow. Structures of ELAC2 in complex with TRMT10C, SDR5C1 and two divergent mt-tRNA substrates reveal two distinct mechanisms of pre-tRNA recognition. While canonical nu-tRNAs and mt-tRNAs are recognized by direct ELAC2–RNA interactions, processing of noncanonical mt-tRNAs depends on protein–protein interactions between ELAC2 and TRMT10C. These results provide the molecular basis for tRNA 3' processing in both the nucleus and the mitochondria and explain the organelle-specific requirement for additional factors. Moreover, they suggest that TRMT10C–SDR5C1 evolved as a mitochondrial tRNA maturation platform to compensate for the structural erosion of mt-tRNAs in bilaterian animals.

Transfer RNAs (tRNAs) are key mediators between the RNA and protein worlds that decode genetic information in mRNAs to sequences of amino acids during translation[1]. They are transcribed as precursor transcripts (pre-tRNAs), which need to be post-transcriptionally processed and modified to become functional[2–5]. The first steps of tRNA maturation involve the removal of leader and trailer sequences at their 5' and 3' ends[5–8]. These universal steps are followed by species-specific and organelle-specific maturation steps, such as 3'-CCA addition, chemical modifications and removal of introns by tRNA splicing[2,4,5,7,9–12]. These steps are crucial for the proper function of tRNAs and mutations in tRNA maturation factors can cause severe diseases in humans[11,13–15].

Consistent with their ancient evolutionary origin and fundamental role in protein biosynthesis, the basic 'canonical' structure of tRNAs is highly conserved[16–18]. They adopt a typical cloverleaf secondary

structure comprising four arms (anticodon arm, acceptor arm, T-arm and D-arm), which fold into an L-shaped tertiary structure[19–21]. Its opposing ends are formed by (1) the acceptor arm, which contains the 5' and 3' ends of the tRNA and serves as the site of aminoacylation, and (2) the anticodon loop, which contains the eponymous anticodon triplet for decoding mRNA codons during ribosomal translation[20,21]. The structural core of canonical tRNAs consists of a tight network of tertiary interactions between the D-loop and T-loop, forming the characteristic 'tRNA elbow' and stabilizing the overall L-shaped fold[21,22]. These conserved structural features serve as recognition sites for enzymes that carry out tRNA maturation and charging[23,24].

In addition to about 300 different cytoplasmic tRNAs encoded by ~500 tRNA genes in the human nuclear genome[25,26], the human mitochondrial genome (mtDNA) encodes a minimal set of 22 tRNAs (mt-tRNAs),

[1]Department of Cellular Biochemistry, University Medical Center Göttingen, Göttingen, Germany. [2]Research Group Structure and Function of Molecular Machines, Max Planck Institute for Multidisciplinary Sciences, Göttingen, Germany. [3]International Max Planck Research School for Molecular Biology, University of Göttingen, Göttingen, Germany. [4]Department of Molecular Biology, University Medical Center Göttingen, Göttingen, Germany. [5]Cluster of Excellence "Multiscale Bioimaging: from Molecular Machines to Networks of Excitable Cells" (MBExC), University of Göttingen, Göttingen, Germany. [6]Research Group Structure and Function of Molecular Machines, Göttingen Center for Molecular Biosciences (GZMB), University of Göttingen, Göttingen, Germany. [7]These authors contributed equally: Arjun Bhatta, Bernhard Kuhle, Ryan D. Yu. ✉e-mail: hauke.hillen@med.uni-goettingen.de

which are required for the synthesis of the 13 mtDNA-encoded essential subunits of the oxidative phosphorylation complexes[27]. Despite their key role in cellular energy homeostasis, animal mt-tRNAs underwent a unique process of sequence and structural erosion, leading to the loss of many of the conserved structural features required for nuclear tRNA (nu-tRNA) processing, modification and aminoacylation[28–30]. Consequently, the enzymes performing these steps are specialized in recognizing only nu-tRNAs or only mt-tRNAs or evolved unique mechanisms to serve tRNAs from both genomes[31–35].

The first steps of tRNA processing are carried out by the endoribonucleases (RNases) P and Z, which cleave the 5′ leader and 3′ trailer sequences from primary pre-tRNA transcripts[4,10,36]. In humans, nu-tRNAs and mt-tRNAs are processed at the 5′ end by two different RNase Ps of distinct architectures and evolutionary origins[32,37,38]. The nu-tRNAs are synthesized in the nucleus by RNA polymerase III, resulting in short pre-nu-tRNA transcripts 80–100 nt in length. Following transcription, the nascent nu-tRNA precursor is bound by La protein, which protects its 3′ terminus against exonucleolytic degradation and functions as an RNA chaperone to facilitate correct tRNA folding[39–41]. Subsequently, pre-nu-tRNAs are processed by a large ribozyme–protein complex (nu-RNase P), which recognizes the acceptor–T-arm domain and the elbow of its pre-tRNA substrate[42]. By contrast, mt-tRNAs are transcribed in the mitochondrial matrix by a dedicated mitochondrial RNA polymerase as part of two long polycistronic transcripts containing all 22 mt-tRNAs interspersed between mt-mRNA and mt-rRNA sequences[27,43,44]. The mt-tRNA 5′ cleavage from these transcripts is carried out by a protein-only RNase P composed of the methyltransferase TRMT10C (MRPP1), the dehydrogenase SDR5C1 (MRPP2/HSD17B10) and the endoribonuclease PRORP (MRPP3)[32], recognizing pre-tRNA substrates by their overall L-shaped structure and conserved elements in the anticodon loop[45]. These interactions are mediated by the TRMT10C–SDR5C1 subcomplex, which has been proposed to act as a maturation platform for multiple steps of mt-tRNA processing[45,46].

In contrast to 5′ end processing, 3′ end processing of human nu-tRNAs and mt-tRNAs is catalyzed by a common RNase Z enzyme, named ELAC2 (refs. 31,47). RNase Z enzymes belong to the β-lactamase family of metal-dependent endonucleases[48–50]. The β-lactamase domain contains the active site, with two catalytic $Zn^{2+}$ ions coordinated by a conserved HXHXDH motif, and an RNase Z-specific sequence insertion called the 'flexible arm', which binds the elbow of the pre-tRNA substrate[49–52]. Two types of RNase Z enzymes exist: a ubiquitous short form (RNase $Z_S$) that contains a single β-lactamase domain that assembles into homodimers and a eukaryote-specific long form (RNase $Z_L$) with two β-lactamase domains that presumably evolved by gene duplication and fusion[53]. ELAC2 is an RNase $Z_L$ that localizes to both the nucleus and the mitochondria[31,54] and its knockout leads to impaired 3′ processing of both nu-tRNAs and mt-tRNAs[55]. ELAC2 by itself appears to be sufficient for nu-tRNA processing, as it can efficiently cleave nu-tRNA precursors without additional factors[56]. By contrast, in vitro studies show that ELAC2 requires TRMT10C and SDR5C1 for efficient processing of most mt-tRNAs, suggesting that human mitochondrial RNase Z may be a multisubunit complex[46]. To date, no structures of ELAC2 or other RNase $Z_L$ enzymes in complex with their substrate are available and the structural and mechanistic basis of eukaryotic tRNA 3′ processing remains elusive. Furthermore, it is not known how ELAC2 specifically recognizes its structurally diverse nu-tRNA and mt-tRNA substrates, why the TRMT10C–SDR5C1 subcomplex is required for processing of most mt-tRNAs and how the strict order of tRNA 5′ processing followed by 3′ processing is ensured.

Here, we use a combination of in vitro biochemical assays and single-particle cryogenic electron microscopy (cryo-EM) to elucidate the mechanism of human tRNA 3′ processing. Using a reconstituted system, we show that TRMT10C–SDR5C1 specifically facilitates the processing of structurally degenerate mt-tRNAs with reduced elbow regions, suggesting that it compensates for the absence of this otherwise conserved element. Cryo-EM structures of ELAC2 in complex with TRMT10C–SDR5C1 and either a canonical (mt-tRNA$^{Gln}$) or degenerate mt-tRNA precursor (mt-tRNA$^{Tyr}$) reveal how ELAC2 binds and accurately processes these structurally divergent tRNAs, providing a mechanistic model for tRNA 3′ end processing in both the nucleus and the mitochondria. The structures show that TRMT10C stabilizes the tertiary fold of degenerate mt-tRNAs and facilitates ELAC2 binding through direct protein–protein interactions, thereby compensating for the loss of protein–RNA interactions with the conserved elbow structure in degenerate mt-tRNAs. Lastly, our data suggest a rationale for the strict sequence of tRNA processing steps. Taken together, these results provide a molecular picture of human tRNA 3′ processing in the nucleus and mitochondria, explain the requirement of TRMT10C and SDR5C1 for mt-tRNA processing and yield insights into the evolution of the TRMT10C–SDR5C1 complex as the mt-tRNA maturation platform.

## Results

### TRMT10C and SDR5C1 enable 3′ processing of degenerate mt-tRNAs

To investigate the mechanism of human tRNA 3′ processing, we first set out to understand how a single ELAC2 enzyme can process structurally highly diverse pre-nu-tRNA and pre-mt-tRNA substrates. We started by re-examining previous data by Reinhard et al. on the differential dependence of ELAC2 on TRMT10C–SDR5C1 for in vitro processing of mt-tRNAs[46] (Fig. 1a–c). In particular, we compared the degree to which ELAC2-mediated cleavage was reported to be dependent on TRMT10C–SDR5C1 (strong, intermediate and no dependence) to the structural properties of the respective mt-tRNAs[28,46,57,58]. This revealed that the four mt-tRNAs exhibiting no dependence on TRMT10C–SDR5C1 are the only mt-tRNAs predicted to form nu-tRNA-like canonical or near-canonical tertiary elbow interactions. By contrast, mt-tRNAs with highly reduced D-loops and T-loops and predicted to lack tertiary elbow interactions consistently showed strong dependence on TRMT10C–SDR5C1. This suggests that the differential dependence of tRNAs on TRMT10C–SDR5C1 for 3′ processing by ELAC2 may be related to the structural properties of the tRNA elbow region.

To test this hypothesis, we reconstituted a 3′ processing system using purified recombinant ELAC2, TRMT10C and SDR5C1 and in vitro transcribed pre-tRNA substrates containing either canonical (nu-tRNA$^{Gly}$ and mt-tRNA$^{Gln}$) or highly degenerated tRNA elbow structures (mt-tRNA$^{Ala}$, mt-tRNA$^{Thr}$ and mt-tRNA$^{Tyr}$) (Extended Data Fig. 1a). As expected, ELAC2 was able to efficiently process nu-tRNA$^{Gly}$ and mt-tRNA$^{Gln}$ without additional factors and was even slightly inhibited by TRMT10C–SDR5C1 (Fig. 1d). By contrast, processing of mt-tRNA$^{Ala}$, mt-tRNA$^{Thr}$ and mt-tRNA$^{Tyr}$ was strictly dependent on the presence of TRMT10C–SDR5C1 (Fig. 1d). The same results were obtained with mt-tRNA$^{Gln}$ transcripts containing the 3′ trailer of pre-mt-tRNA$^{Tyr}$, confirming that the observed differences in TRMT10C–SDR5C1 dependence are only due to the properties of the tRNAs themselves and due to their 3′ trailer sequences (Extended Data Fig. 1b). Consistent with these results, fluorescence anisotropy (FA) experiments with FAM-labeled tRNA precursors revealed that ELAC2 has markedly higher affinity for canonical than for noncanonical pre-tRNAs ($K_d = 301$ nM for mt-tRNA$^{Gln}$ versus $K_d > 6$ μM for mt-tRNA$^{Tyr}$), while the TRMT10C–SDR5C1 complex shows comparable affinities for both ($K_d = 167$ nM and 215 nM for mt-tRNA$^{Gln}$ and mt-tRNA$^{Tyr}$, respectively) (Extended Data Fig. 1c). Moreover, the differential dependence on TRMT10C–SDR5C1 was only observed for 3′ processing by ELAC2, whereas PRORP requires TRMT10C–SDR5C1 for 5′ processing of both mt-tRNA$^{Gln}$ and mt-tRNA$^{Tyr}$ precursors (Extended Data Fig. 1d).

To confirm that the differential TRMT10C–SDR5C1 dependence of mt-tRNA 3′ processing indeed relates specifically to the degeneracy of the tRNA elbow, we generated three mutants of mt-tRNA$^{Gln}$ containing either a G19C single substitution, a G18C;G19C double substitution or a transplantation of the T-loop from mt-tRNA$^{Tyr}$ (denoted T-loop$^{Tyr}$),

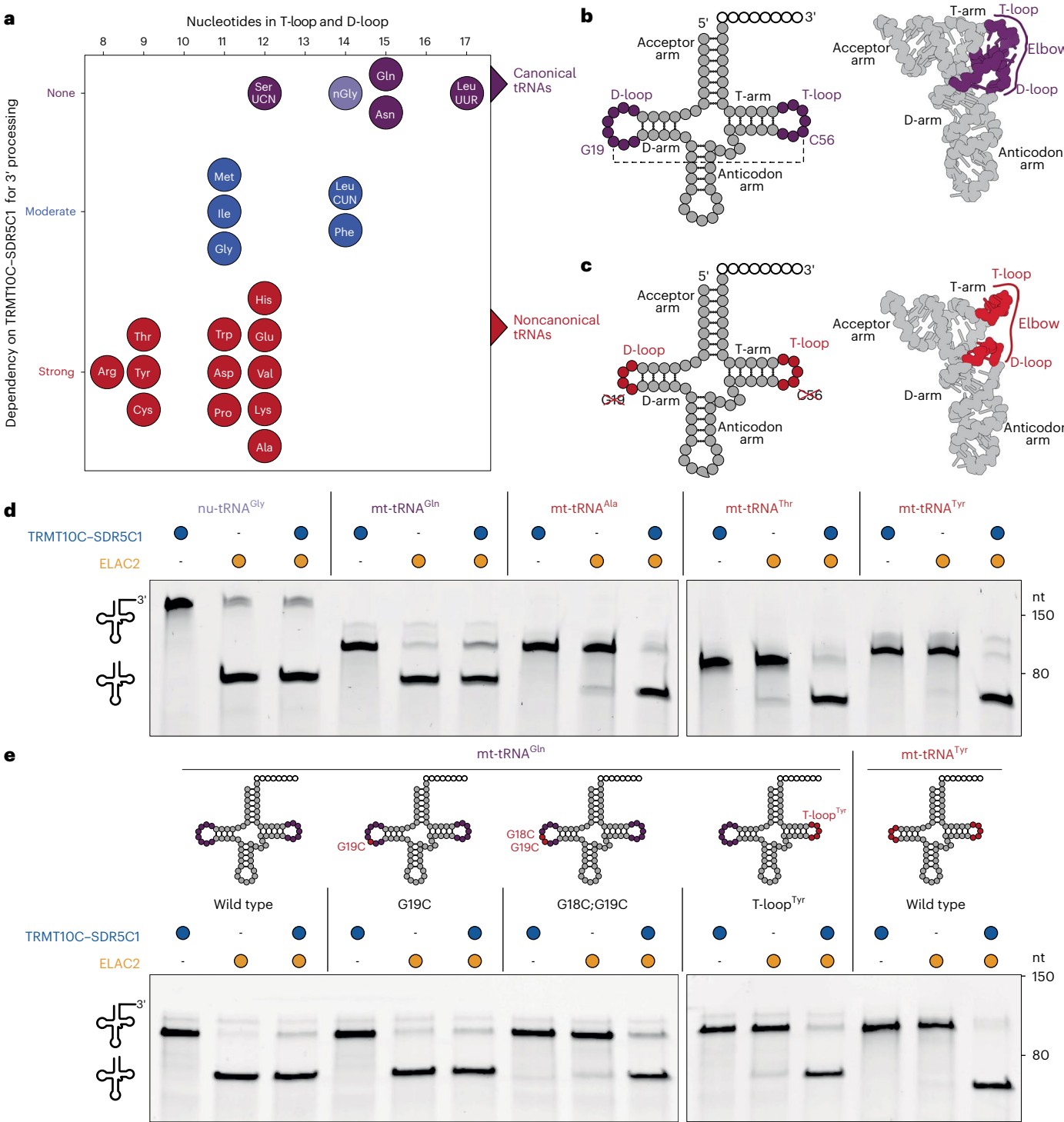

**Fig. 1 | Differential dependence of nu-tRNAs and mt-tRNAs on TRMT10C–SDR5C1 for 3′ processing by ELAC2. a,** The dependency of mt-tRNAs on TRMT10C–SDR5C1 for 3′ processing by ELAC2 correlates with the degeneracy of their elbow region. mt-tRNAs are labeled with the three-letter codes for their cognate amino acids. mt-tRNA[Ser(AGY)] is not included, as it is not processed efficiently by ELAC2 in the presence or absence of TRMT10C–SDR5C1 (refs. [46],[74]). The sum of nucleotides in the D-loops and T-loops of each mt-tRNA is plotted against their dependency on TRMT10C–SDR5C1 according to Reinhard et al.[46]. For reference, the human nu-tRNA[Gly] is included (nGly), which shows no dependence on TRMT10C–SDR5C1 (**d**). **b,** Secondary cloverleaf fold (left) and tertiary structural schematic (right) of a canonical tRNA. The D-loop and T-loop

regions, which interact to form the canonical elbow, are highlighted in purple. **c,** Secondary cloverleaf fold (left) and tertiary structural schematic (right) of a noncanonical mt-tRNA. The D-loop and T-loop regions are highlighted in red. Unlike canonical tRNAs, noncanonical tRNAs lack the conserved D-loop and T-loop nucleotides that form tertiary elbow interactions in canonical tRNAs. **d,** In vitro cleavage assays showing the differential dependency between canonical and degenerated tRNAs on TRMT10C–SDR5C1 for 3′ processing by ELAC2. **e,** In vitro cleavage assays demonstrating the gain of TRMT10C–SDR5C1 dependence by mt-tRNA[Gln] upon the introduction of mutations disrupting the canonical elbow structure. Gels in **d,e** are representatives of three independent replicates.

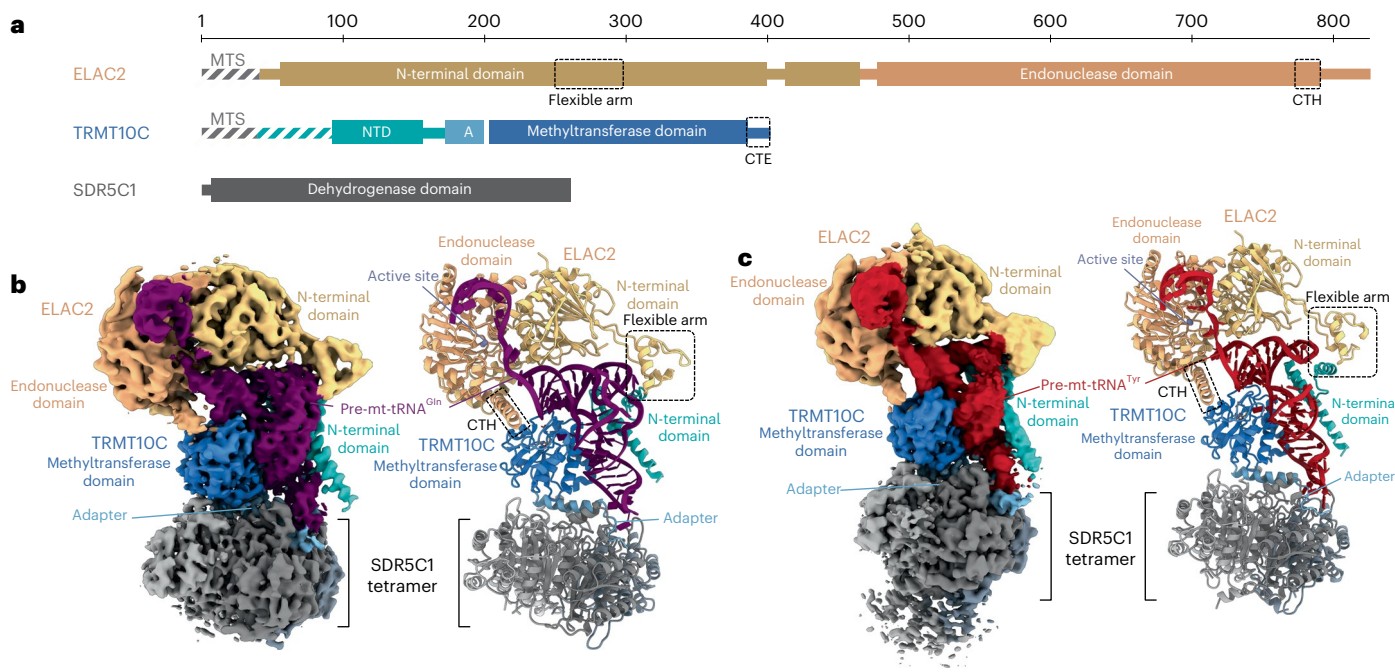

**Fig. 2 | Structures of mitochondrial RNase Z complexes. a**, Domain representation of mt-RNase Z subunits. ELAC2 domains are shown in shades of orange, TRMT10C domains are shown in shades of blue and the four SDR5C1 monomers are shown in shades of gray throughout the paper. Mitochondrial targeting sequences of ELAC2 and TRMT10C are labelled as MTS. Marked by dashed boxes are the ELAC2 flexible arm and CTH and the TRMT10C CTE. Regions of the wild-type proteins lacking in the recombinant constructs used here are marked with striped lines. Mitochondrial targeting signals of ELAC2 and TRMT10C are indicated. **b**, Cryo-EM density map (left) and cartoon representation of the structural model (right) of the mt-RNase Z[Gln] complex. The two catalytic Zn[2+] ions in the active site are shown as gray spheres. The pre-mt-tRNA[Gln] is colored in purple. The ELAC2 flexible arm and CTH are marked with black dashed rectangles. **c**, Cryo-EM density map (left) and cartoon representation of the structural model (right) of the mt-RNase Z[Tyr] complex. The catalytic Zn[2+] ions in the active site are shown as gray spheres. The pre-mt-tRNA[Tyr] is colored in red. The ELAC2 flexible arm and CTH are marked with black dashed rectangles.

leading to a weak, moderate and strong disruption of the canonical elbow, respectively. While G19C had virtually no effect on 3′ processing by ELAC2 in the absence of TRMT10C–SDR5C1, G18C;G19C and T-loop[Tyr] nearly completely abolished 3′ processing by ELAC2 alone (Fig. 1e). Importantly, in all cases, these processing defects could be rescued by the addition of TRMT10C–SDR5C1. Taken together, these results demonstrate that the dependence of ELAC2 on additional protein factors for 3′ processing is related to the structural degeneracy of mt-tRNAs and that TRMT10C–SDR5C1 compensates for the absence of canonical tRNA elbow structures.

**Structures of ELAC2 with TRMT10C–SDR5C1 and mt-tRNAs**

To determine the molecular basis of tRNA recognition and 3′ processing by ELAC2, we reconstituted RNase Z with either the canonical pre-mt-tRNA[Gln] (mt-RNase Z[Gln]) or the highly degenerated pre-mt-tRNA[Tyr] substrate (mt-RNase Z[Tyr]) and a catalytically inactive variant of ELAC2 harboring an Asp550Asn substitution (ELAC2[mut])[59] (Extended Data Fig. 2). The mt-RNase Z[Tyr] complex was reconstituted by first incubating a 5′ leaderless pre-mt-tRNA[Tyr] substrate with TRMT10C–SDR5C1, followed by the addition of ELAC2[mut]. As ELAC2 does not require TRMT10C–SDR5C1 for processing of pre-mt-tRNA[Gln], it was not clear whether these subunits would also be part of the mt-RNase Z[Gln] complex. We, therefore, recapitulated the physiological order of events by first processing a pre-mt-tRNA[Gln] substrate containing a 5′ leader by mt-RNase P, followed by the subsequent addition of ELAC2[mut]. Although TRMT10C–SDR5C1 is required only for 5′ processing but not 3′ processing of mt-tRNA[Gln], the pre-mt-tRNA[Gln] remained stably bound to TRMT10C–SDR5C1 following PRORP release and association of ELAC2[mut], suggesting that the TRMT10C–SDR5C1 complex also forms

part of the mt-RNase Z complex for canonical mt-tRNAs (Extended Data Fig. 2d).

We then determined the structures of mt-RNase Z[Gln] and mt-RNase Z[Tyr] using single-particle cryo-EM at global resolutions of 3.4 and 3.2 Å, with the ELAC2 region at 4.0 and 3.6 Å after focused refinement, respectively (Extended Data Figs. 3 and 4). From the mt-RNase Z[Tyr] dataset, the ELAC2 core was further resolved at a higher resolution of 3.2 Å (Extended Data Figs. 3 and 4). This allowed us to fit and remodel the structures of TRMT10C, the SDR5C1 tetramer and pre-tRNA[Tyr] on the basis of the previously determined mt-RNase P complex structure (mt-RNase P[Tyr]) and the AlphaFold model of ELAC2 (ref. 60), resulting in complete structural models of the mt-RNase Z[Tyr] and mt-RNase Z[Gln] complexes (Fig. 2 and Table 1).

Both mt-RNase Z complexes share the same architecture, which resembles that of mt-RNase P[Tyr] (ref. 45). The SDR5C1 tetramer forms a base to which TRMT10C is anchored through its 'adapter helix' and, together, they form a platform that binds the pre-tRNA substrate through interactions with all four tRNA subdomains. ELAC2 binds on top of the pre-tRNA and occupies the same position as PRORP in the RNase P complex, showing that binding of these two endoribonucleases is mutually exclusive[45]. ELAC2 folds into two β-lactamase domains (N-terminal domain (NTD) and endonuclease domain), which form extensive interactions with the pre-tRNA through the acceptor stem, T-arm and 3′ trailer. Additional contacts with the T-loop and elbow region are formed by the ~50-residue-long flexible arm insertion in the ELAC2 NTD. ELAC2 also contacts TRMT10C through its flexible arm and a C-terminal helix (CTH). In both mt-RNase Z complexes, ELAC2 exhibits conformational variability with respect to the TRMT10C–SDR5C1–pre-tRNA subcomplex (Extended Data Fig. 5a,b).

**Table 1 | Cryo-EM data collection, refinement and validation statistics**

| | mt-RNase Z$^{Gln}$ complex (EMD-19455) (PDB 8RR3) | mt-RNase Z$^{Tyr}$ complex (EMD-19457) (PDB 8RR4) |
|---|---|---|
| **Data collection and processing** | | |
| Magnification | ×105,000 | ×105,000 |
| Acceleration voltage (kV) | 300 | 300 |
| Electron exposure (e⁻ per Å²) | 42.0 | 39.6 |
| Nominal defocus range (µm) | −0.5 to −2.5 | −0.5 to −2.5 |
| Pixel size (Å) | 0.834 | 0.834 |
| Symmetry imposed | *C1* | *C1* |
| Initial particle images (no.) | 12,168,828 | 5,987,454 |
| Final particle images (no.) | 75,812 | 28,023 |
| Map resolution (Å) | 3.4 | 3.2 |
| FSC threshold | 0.143 | 0.143 |
| Map resolution range (Å) | 9–2.9 | 11–2.8 |
| **Map sharpening *B* factors (Å²)** | | |
| Consensus map | −47.4 | −56.6 |
| Composite map | 0 | 0 |
| **Refinement** | | |
| Initial models (PDB) | 7ONU | 7ONU |
| Initial models (AlphaFoldDB) | Q9BQ52 | Q9BQ52 |
| Model resolution (Å) | 3.5 | 3.4 |
| FSC threshold | 0.5 | 0.5 |
| Model composition | | |
| Nonhydrogen atoms | 16,294 | 16,539 |
| Protein/nucleotide residues | 1,915/81 | 1,961/76 |
| Ligands | Zn: 2, SAH: 1 | Zn: 2, SAH: 1 |
| ***B* factors (Å²), min/max/mean** | | |
| Protein | 30/312/130 | 33/367/154 |
| Nucleotide | 72/512/170 | 61/531/196 |
| Ligand | 275/309/292 | 356/382/369 |
| R.m.s.d. | | |
| Bond lengths (Å) | 0.005 | 0.003 |
| Bond angles (°) | 1.166 | 0.602 |
| **Validation** | | |
| MolProbity score | 1.74 | 1.50 |
| Clashscore | 10.96 | 7.38 |
| Rotamer outliers (%) | 0.58 | 0.5 |
| Ramachandran plot | | |
| Favored (%) | 96.93 | 97.52 |
| Allowed (%) | 2.97 | 2.43 |
| Outliers (%) | 0.11 | 0.05 |

FSC, Fourier shell correlation.

The RNase Z-bound mt-tRNA$^{Gln}$ and mt-tRNA$^{Tyr}$ adopt similar L-shaped overall folds but also exhibit important structural differences between the two complexes. First, mt-tRNA$^{Gln}$ and mt-tRNA$^{Tyr}$ adopt distinct anticodon stem-loop topologies, which are recognized by TRMT10C through different sets of interactions (Extended Data Fig. 5c,d). Thus, the mechanism of pre-tRNA anticodon loop recognition by TRMT10C–SDR5C1 appears more versatile than previously suggested on the basis of the mt-RNase P$^{Tyr}$ structure (Extended Data Fig. 5e,f)[45]. Second, while pre-mt-tRNA$^{Gln}$ adopts a stable canonical elbow structure, no stable D-loop–T-loop tertiary interactions are formed in pre-mt-tRNA$^{Tyr}$. Consequently, the mt-RNase Z$^{Gln}$ and mt-RNase Z$^{Tyr}$ complexes differ with respect to their specific protein–RNA and protein–protein interactions.

## Recognition of canonical tRNAs by ELAC2

The structure of mt-RNase Z$^{Gln}$ reveals how ELAC2 recognizes and interacts with canonical pre-tRNA substrates. The pre-mt-tRNA$^{Gln}$ adopts a compact L-shaped structure, from which the 3′ trailer extends as a four nucleotides long single-stranded region, followed by a short stem loop (Fig. 3a,b). The T-loop of mt-tRNA$^{Gln}$ adopts a characteristic pentanucleotide U-turn structure[61], which establishes mutually stabilizing tertiary interactions with nucleobases G18 and G19 from the D-loop to form the canonical tRNA elbow (Fig. 3c,d). This includes a G19–C56 Watson–Crick base pair at the distal end of the elbow, which represents one of the most highly conserved features of canonical tRNAs[22]. These interactions in the mt-tRNA$^{Gln}$ elbow are stable despite a large TRMT10C-induced distortion of the D-stem of up to 10 Å compared to free tRNA[62] (Extended Data Fig. 6a).

ELAC2 clamps the acceptor arm–T-arm 'minihelix' of pre-tRNA$^{Gln}$ between its NTD and endonuclease domain (Fig. 3b). The ELAC2 NTD and flexible arm form extensive contacts along the T-stem and with several conserved nucleobases of the T-loop, while the acceptor arm is inserted between the ELAC2 endonuclease domain and CTH. Both interfaces are mediated primarily by electrostatic interactions between basic protein side chains and the ribose phosphate backbone of the tRNA (Fig. 3e). At the distal end of the tRNA elbow, a hydrophobic patch (residues 268–280 in helix α4) in the globular subdomain of the ELAC2 flexible arm interacts with the conserved G19–C56 tertiary base pair, stacking against its extended aromatic ring system (Fig. 3d and Extended Data Fig. 6b). This interaction appears to be relatively stable, as the flexible arm shows lower conformational variability and atomic *B* factors compared to other parts of ELAC2 (Extended Data Fig. 6c). This suggests that the canonical tRNA elbow structure serves as an important determinant for TRMT10C–SDR5C1-independent recognition of canonical mt-tRNAs by ELAC2, consistent with our biochemical data. ELAC2 further interacts with the 3′ trailer RNA through a positively charged patch in the endonuclease domain (Extended Data Fig. 6d), although the resolution is not sufficient to resolve specific interactions. Together, these interactions position the 3′ end of the tRNA near the endonuclease active site of ELAC2.

Although TRMT10C–SDR5C1 is not required for efficient processing of pre-mt-tRNA$^{Gln}$, ELAC2 also interacts with TRMT10C through two interfaces in the mt-RNase Z$^{Gln}$ complex. The first is formed near the tRNA elbow and involves hydrophobic interactions between the ELAC2 flexible arm and the TRMT10C NTD (Fig. 3f and Extended Data Fig. 6e). This interface is 1.9-fold smaller than the adjacent interface between the flexible arm and the tRNA (158 Å² between flexible arm and TRMT10C NTD versus 308 Å² between flexible arm and tRNA), suggesting that the latter interface has the primary role in ELAC2 binding to mt-tRNA$^{Gln}$. Consistent with this, deletion of the ELAC2 flexible arm (Δ250–298) but not TRMT10C NTD (Δ1–124) abolished the 3′ processing of mt-tRNA$^{Gln}$ (Fig. 3g). This shows that stabilization of the flexible arm is crucial for productive substrate binding by ELAC2 and that this is achieved primarily through interactions with the tRNA elbow rather than with the TRMT10C NTD. The second interface is formed between the ELAC2 CTH and the TRMT10C methyltransferase domain (MTD) and C-terminal extension (CTE). This interface is only observable at a low map threshold, indicating a dynamic interaction interface (Extended Data Fig. 6f). While deletion of the ELAC2 CTH (Δ772–826) reduced 3′ processing of mt-tRNA$^{Gln}$, deletion of the TRMT10C CTE (Δ385–403) showed no effect (Fig. 3g). Thus, this interface also does not seem to

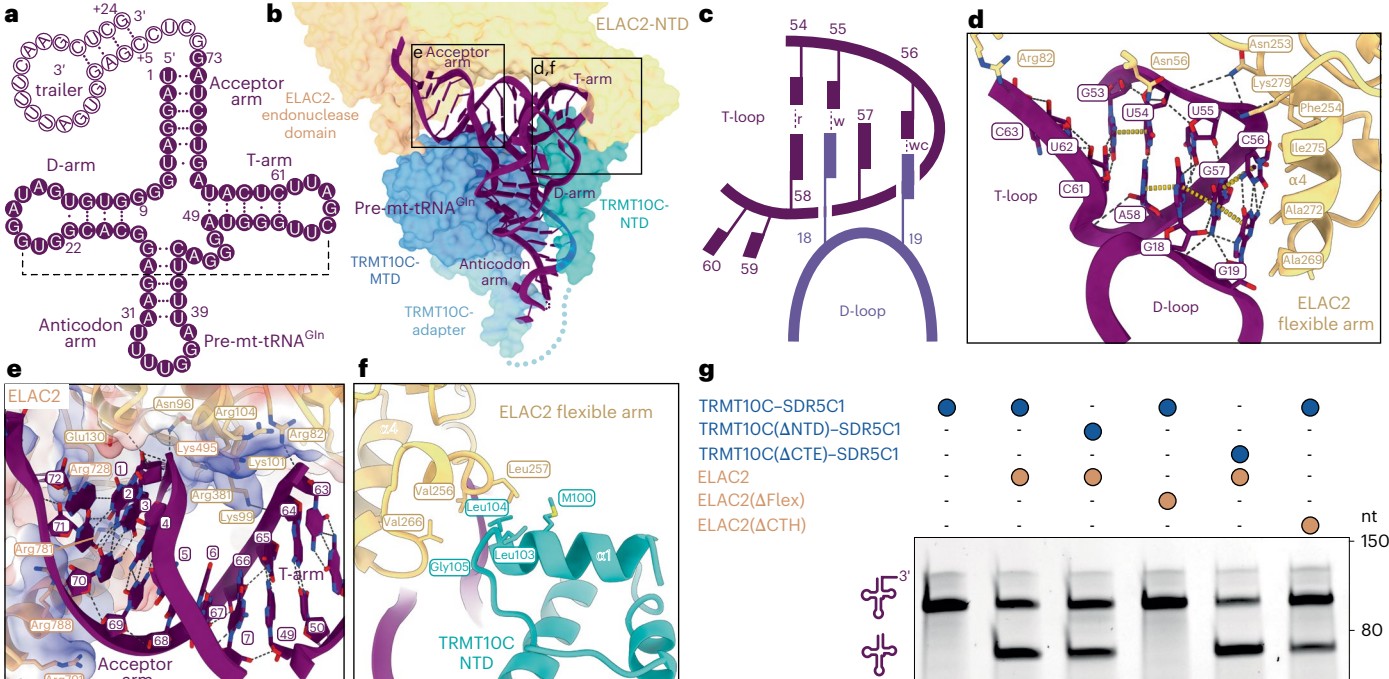

**Fig. 3 | Mechanism of canonical mt-tRNA recognition by ELAC2. a**, Secondary-structure representation of pre-mt-tRNA$^{Gln}$. The tRNA subdomains and the 3′ trailer are indicated. Nucleotides of the mature tRNA are shown as solid circles; 3′ trailer nucleotides are shown as hollow circles. G·C, A·U and G·U base pairs are marked with three, two or one dot, respectively. The tertiary G19·C56 base pair is indicated by a black dashed line. **b**, Overview of the structure and interactions of pre-mt-tRNA$^{Gln}$ in the mt-RNase Z$^{Gln}$ complex. ELAC2 and TRMT10C are shown as transparent surfaces and the pre-mt-tRNA$^{Gln}$ is shown in cartoon representation. Regions shown in detail in **d**–**f** are indicated. **c**, Schematic representation of the tertiary interactions between the D-loop and T-loop in the mt-tRNA$^{Gln}$ elbow. Nucleobases are represented as rectangular blocks connected to the ribose phosphate backbone represented as solid curved lines. Watson–Crick (wc), wobble (w) and reverse Hoogsteen (r) base pairs are indicated as dashed lines. **d**, Recognition of the mt-tRNA$^{Gln}$ elbow by ELAC2 flexible arm. ELAC2 flexible

arm side chains within 4.5 Å of the tRNA are shown as sticks. Hydrogen bonds and stacking interactions are shown as black dashed and yellow dotted lines, respectively. Helix α4 of ELAC2, which forms hydrophobic interactions with the G19–C56 base pair, is marked. **e**, Interactions of ELAC2 with the pre-mt-tRNA$^{Gln}$ backbone in the acceptor and T-arm. The surface electrostatic potential of ELAC2 is shown transparently as a three-color gradient scheme from −24 to +15 kcal mol$^{-1}$ per e$^-$ (red, negative; white, neutral; blue, positive). ELAC2 side chains within 4.5 Å of the tRNA are shown as sticks. **f**, Interactions of the ELAC2 flexible arm with the TRMT10C NTD. Side chains within 4.5 Å of the interface are shown as sticks. Helix α4 of ELAC2 and helix α1 of TRMT10C are labeled. **g**, In vitro cleavage assays showing the effect of ELAC2 and TRMT10C deletion mutants on 3′ processing of pre-mt-tRNA$^{Gln}$. Representative gel of three independent replicates.

have a crucial role in canonical pre-tRNA recognition and processing. Taken together, these data show that ELAC2 interacts with canonical tRNAs primarily through direct protein–RNA interactions with the acceptor stem, T-arm and elbow region, while protein–protein interactions with TRMT10C are not strictly required.

Our structural and biochemical results also suggest a conserved substrate recognition mechanism for canonical mt-tRNAs and nu-tRNAs (Fig. 1). A structural comparison of free yeast nu-tRNA$^{Phe}$ and mt-tRNA$^{Gln}$ in the mt-RNase Z$^{Gln}$ complex showed that the tRNA structural elements recognized by ELAC2 remain unperturbed between free and TRMT10C–SDR5C1-bound tRNAs[62] (Extended Data Fig. 6a). Therefore, the structure of mt-RNase Z$^{Gln}$ enables us to construct a model of substrate-engaged nuclear RNase Z by superimposing human nu-tRNA$^{Gly}$ onto the complex and omitting TRMT10C–SDR5C1. To further verify this model, we predicted the structure of the ELAC2–nu-tRNA$^{Gly}$ complex using AlphaFold3 (ref. 63), which resulted in a highly similar structure (Extended Data Fig. 6g). In the model, all ELAC2–RNA interactions observed in mt-RNase Z$^{Gln}$, particularly those of the flexible arm with the canonical elbow structure, could be established on nu-tRNA without major structural rearrangements, even in the absence of TRMT10C–SDR5C1 (Fig. 4b). This suggests that the canonical elbow structure also serves as major determinant for nu-tRNA recognition by ELAC2. Consistent with this, mutational disruption of the elbow in nu-tRNA$^{Gly}$ by transplantation of the T-loop from mt-tRNA$^{Tyr}$ resulted in the complete loss of 3′ end processing, while the less disruptive

G18C;G19C substitution did not notably affect nu-tRNA$^{Gly}$ processing (Fig. 4c). The processing defect in pre-nu-tRNA$^{Gly}$(T-loop$^{Tyr}$) was rescued by the addition of TRMT10C–SDR5C1, demonstrating that disruption of the canonical elbow in nu-tRNAs leads to the same dependency on TRMT10C–SDR5C1 as observed for degenerated mt-tRNAs.

In summary, our data show how ELAC2 recognizes and interacts with canonical tRNAs both in the mitochondria and in the nucleus.

**Recognition of degenerate mt-tRNAs by TRMT10C and ELAC2**

The structure of mt-RNase Z$^{Tyr}$ reveals how TRMT10C and SDR5C1 facilitate processing of structurally degenerate mt-tRNAs (Fig. 5). Overall, interactions between ELAC2 and pre-tRNA$^{Tyr}$ involve largely the same ribose phosphate interactions in the acceptor–T-arm minihelix and the 3′ trailer as observed in mt-RNase Z$^{Gln}$ (Fig. 5a,b,e and Extended Data Fig. 7a–c). However, the tRNA elbow structure that is recognized by ELAC2 in canonical tRNAs is absent in mt-tRNA$^{Tyr}$, as no stable tertiary interactions are formed between the D-loop and T-loop (Fig. 5c,d and Extended Data Fig. 7d). This results in a higher structural flexibility in the elbow region of pre-mt-tRNA$^{Tyr}$ compared to pre-mt-tRNA$^{Gln}$, with the D-loop and A56 and A57 at the tip of the T-loop poorly resolved in the EM density. Consequently, the extensive interactions between the ELAC2 flexible arm and the elbow observed in mt-RNase Z$^{Gln}$ are absent in mt-RNase Z$^{Tyr}$, resulting in a threefold smaller protein–RNA interface (98 Å$^2$ buried surface area in mt-RNase Z$^{Tyr}$ versus 308 Å$^2$ in mt-RNase Z$^{Gln}$). By contrast, the interaction interface between the ELAC2

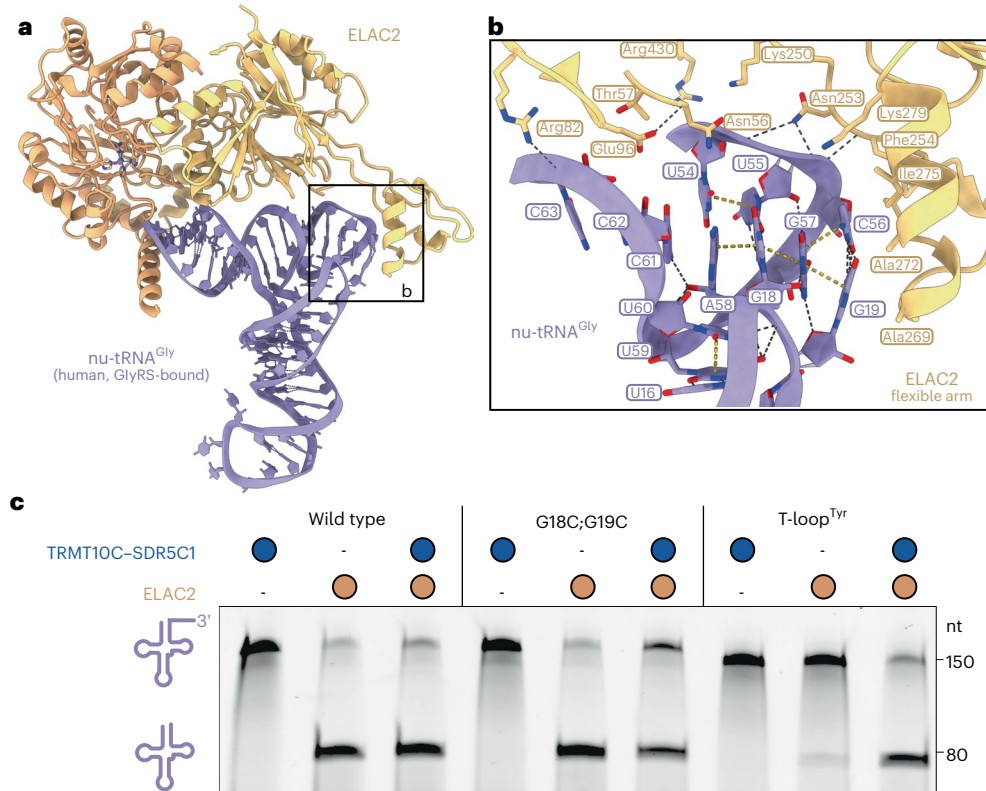

**Fig. 4 | nu-tRNA 3′ processing by ELAC2. a**, Model of nu-tRNA bound to ELAC2. Human nu-tRNA^Gly (bound to the glycyl-tRNA synthetase; PDB 5E6M)[84] (light purple) was superimposed through its acceptor and T-arms to mt-tRNA^Gln in the mt-RNase Z complex. TRMT10C and SDR5C1 are omitted. The region shown in **b** is indicated. **b**, Model for nu-tRNA^Gly elbow recognition by the ELAC2 flexible arm. ELAC2 side chains within 4.5 Å of the tRNA are shown as sticks. **c**, In vitro cleavage assays demonstrating the critical role of the elbow region in nu-tRNA^Gly for 3′ processing by ELAC2. Representative gel of three independent replicates.

flexible arm and the TRMT10C NTD is 1.9-fold larger in mt-RNase Z^Tyr than in mt-RNase Z^Gln (295 Å² in mt-RNase Z^Tyr versus 158 Å² in mt-RNase Z^Gln). This larger protein–protein interface in mt-RNase Z^Tyr is possible because of a reorientation of the ELAC2 flexible arm and TRMT10C NTD toward each other, which positions the TRMT10C NTD directly underneath the globular subdomain of the flexible arm (Fig. 5f and Extended Data Fig. 7e). This predominantly hydrophobic interface involves Met100, Leu103 and Leu104 of TRMT10C and Val256, Leu257, Lys260, Val266 and Gly267 of ELAC2 and may be further stabilized by electrostatic interactions between ELAC2 Lys260 and TRMT10C Glu96 and/or Glu99. As in mt-RNase Z^Gln, a second conformationally dynamic interface is likely formed in mt-RNase Z^Tyr between the CTH of ELAC2 and the MTD and CTE of TRMT10C, which may involve an electrostatic interaction between ELAC2 Arg791 and TRMT10C Asp339 (Extended Data Fig. 7c).

To determine the role of the two ELAC2–TRMT10C interfaces in the processing of noncanonical mt-tRNA substrates, we tested deletion mutants of each of the involved domains in our in vitro cleavage assay (Fig. 5g). The most severe effect was observed for the ELAC2 flexible arm deletion, which abolished 3′ processing of pre-mt-tRNA^Tyr, similar to pre-mt-tRNA^Gln (Figs. 3g and 5g). However, in contrast to pre-mt-tRNA^Gln, deletion of the NTD of TRMT10C also abolished TRMT10C–SDR5C1-supported 3′ processing of pre-mt-tRNA^Tyr. This demonstrates that the TRMT10C NTD, while not required for canonical tRNAs, is critical for TRMT10C–SDR5C1-dependent 3′ processing by ELAC2 on degenerated mt-tRNA substrates. A less pronounced activity loss was observed for the ELAC2 CTH deletion. Interestingly, this effect was exacerbated on mt-tRNA^Gln variants with a mutationally disrupted elbow region (Extended Data Fig. 7f), suggesting that the CTH may facilitate processing of destabilized tRNAs by tethering ELAC2 to the TRMT10C–SDR5C1–pre-tRNA complex. This effect is likely mediated by direct interactions of the ELAC2 CTH with the tRNA acceptor arm and TRMT10C MTD but not with the TRMT10C CTE, as deletion of the latter had no negative effect on ELAC2-mediated 3′ cleavage (Fig. 5g).

Taken together, our structural and biochemical data demonstrate that protein–protein interactions between ELAC2 and TRMT10C compensate for the lack of extensive protein–RNA interactions with otherwise conserved structural elements in degenerate mt-tRNAs.

## Active-site organization of ELAC2

The active-site organization in human ELAC2 is similar to that of *Saccharomyces cerevisiae* (*Sce*) RNase Z_L (RNase Z_L) or *Bacillus subtilis* (*Bsu*) RNase Z[49,64]. In both mt-RNase Z complexes, the active site contains clear density for two putative Zn²⁺ ions (Extended Data Fig. 8a,b). Zn²⁺_A is coordinated by His546, His548, His644 and Asp666 while Zn²⁺_B is coordinated by His551, Asp666 and His724. The residue substituted in our catalytically inactive variant of ELAC2, Asp550Asn, lies adjacent to the active site. Although Asn550 does not coordinate Zn²⁺_B in our structures, Asp550 likely forms part of the coordination sphere of Zn²⁺_B in wild-type ELAC2. However, structural comparisons to *Sce*-RNase Z_L and *Bsu*-RNase Z showed that the Asp550Asn substitution does not substantially alter the active-site architecture or relative positioning of catalytic residues and ions (root-mean-square deviation (r.m.s.d.) at active site: *Sce*-RNase Z_L, 0.84 Å; *Bsu*-RNase Z, 0.63 Å) (Extended Data Fig. 8c).

Because of the large conformational variability of ELAC2 with respect to TRMT10C–SDR5C1–pre-tRNA, the pre-tRNA adopts an ensemble of 'productive' and 'nonproductive' conformations near the 3′ cleavage site. To better resolve the productive active-site conformation, we performed a three-dimensional (3D) variability analysis of mt-RNase Z^Tyr particles with respect to the ELAC2 conformational state (Extended Data Fig. 8d). This resulted in a class where the

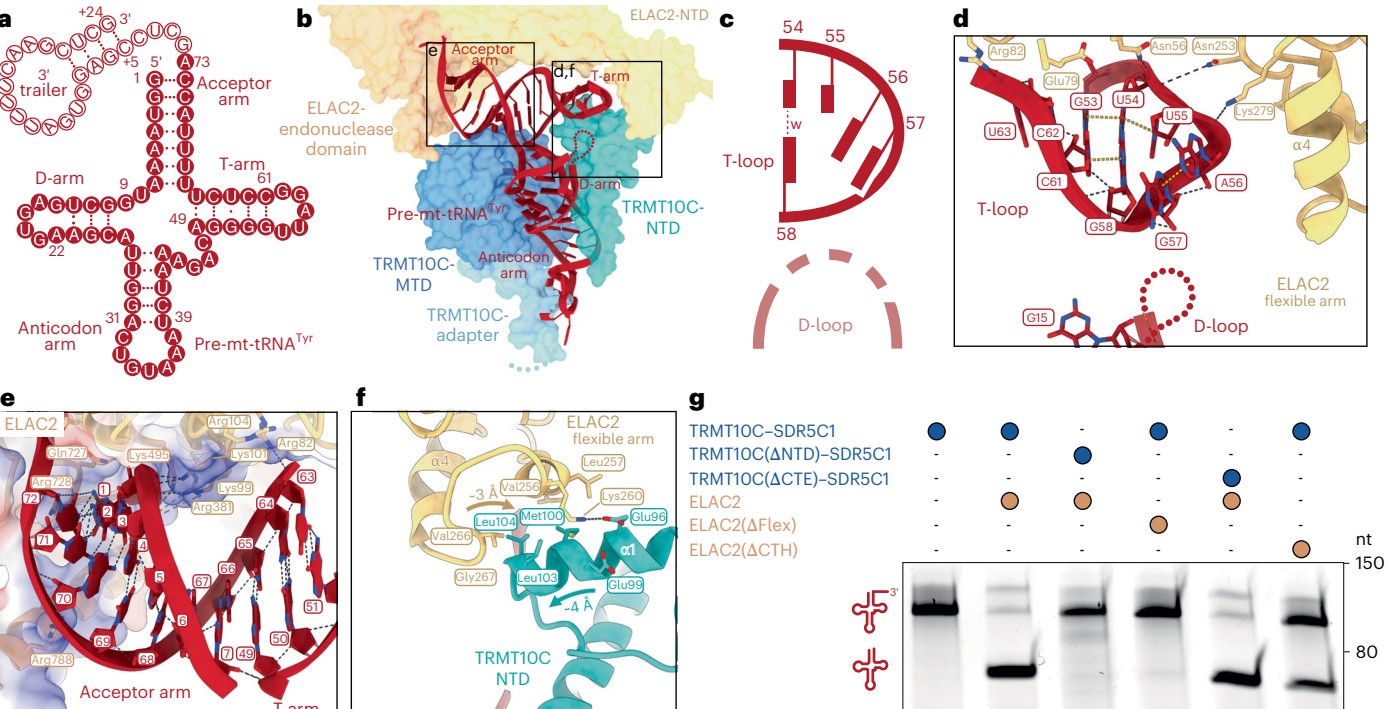

**Fig. 5 | TRMT10C and SDR5C1 facilitate recognition of noncanonical mt-tRNAs by ELAC2. a**, Secondary-structure representation of pre-mt-tRNA^Tyr. Nucleotides of the mature tRNA are shown as solid circles; the 3′ trailer is shown as open circles. **b**, Overview of the structure and interactions of pre-mt-tRNA^Tyr in the mt-RNase Z^Tyr complex. Representation as in Fig. 3b. Regions shown in detail in **d**–**f** are indicated. **c**, Schematic representation of the tertiary interactions between the D-loop and T-loop in the mt-tRNA^Tyr elbow region. Representation as in Fig. 3c. **d**, Interactions between mt-tRNA^Tyr elbow and ELAC2 flexible arm. ELAC2 flexible arm side chains within 4.5 Å of the tRNA are shown as sticks. **e**, Interactions of

ELAC2 with the pre-mt-tRNA^Tyr backbone in the acceptor arm and T-arm. The surface electrostatic potential of ELAC2 is shown transparently as a three-color gradient scheme from −18 to +17 kcal mol⁻¹ per e⁻ (red, negative; white, neutral; blue, positive). ELAC2 side chains within 4.5 Å of the tRNA are shown as sticks. **f**, Interactions of ELAC2 flexible arm with TRMT10C NTD. Side chains within 4.5 Å of the interface are shown as sticks. Helix α4 of ELAC2 and helix α1 of TRMT10C are labeled. **g**, In vitro cleavage assays demonstrating the roles of ELAC2 flexible arm and TRMT10C NTD in 3′ processing of pre-mt-tRNA^Tyr. Representative gel of three independent replicates.

pre-mt-tRNA^Tyr adopts a 'contracted' conformation, allowing the RNA to be positioned in the ELAC2 active site. Although the resulting maps are of limited resolution (~4.1 Å), comparison to the pre-tRNA-bound structure of the *Bsu*-RNase Z homodimer allowed us place the RNA in the active site of ELAC2 such that the scissile phosphodiester is positioned next to the Zn²⁺ ions[49,65] (Extended Data Fig. 8e,f). This model shows that Asp550, proposed to act as the general base catalyst[49,66], and His702, proposed to protonate the leaving group[66], are positioned near the scissile phosphate. The discriminator nucleotide (A73) is likely stabilized by Lys700, while the nucleobase in position 75 is stabilized in a groove formed by residues Leu547, His548, Cys645, Lys646, His647 and Asn583. Both interactions may contribute to positioning the scissile phosphodiester bond in the active site.

In summary, our structural data provide a precatalytic model of substrate-engaged ELAC2, which suggests a conserved catalytic mechanism of RNase Z enzymes.

### Structural basis for sequential tRNA processing

For most human mt-tRNAs and nu-tRNAs, maturation proceeds sequentially, starting with 5′ end processing by RNase P followed by 3′ end processing by ELAC2 (refs. 47,67,68). Superimposing mt-RNase P or nu-RNase P complexes with RNase Z reveals steric clashes between PRORP or nu-RNase P and ELAC2. This suggests that association of 5′ and 3′ processing enzymes to the pre-tRNA is mutually exclusive (Extended Data Fig. 9a,b). Furthermore, the 5′ phosphate of the ELAC2-bound pre-tRNA is displaced by ~6 Å compared to the mt-RNase P-bound state[45] and is buried near the ELAC2 core such that binding of a 5′ unprocessed pre-tRNA would be disfavored because of steric clashes between the 5′ leader and ELAC2. In particular, a loop between residues 726 and 731

and the CTH of ELAC2 would clash with the 5′ leader at positions −2 and −3, respectively (Extended Data Fig. 9c,d). Deletion of the ELAC2 CTH alone does not abolish the discrimination against pre-tRNA with a 5′ leader by ELAC2, suggesting that multiple elements of ELAC2 are involved in ensuring hierarchical processing (Extended Data Fig. 9e).

In conclusion, we found that sequential tRNA 5′ and 3′ end processing requires an exchange of RNase P and RNase Z enzymes and this strict processing order is likely ensured by steric discrimination against 5′ unprocessed pre-tRNAs by ELAC2.

## Discussion

Here, we present structures of human mt-RNase Z bound to two structurally divergent pre-tRNA substrates, which serve as models for both mt-tRNA and nu-tRNA 3′ processing. Together with biochemical data, the structures reveal the molecular determinants for substrate recognition by RNase Z and explain why ELAC2 requires TRMT10C–SDR5C1 to process most mt-tRNAs. Furthermore, they provide insight into the active-site arrangement of RNase Z and enable us to propose a molecular model for sequential processing of nu-tRNAs and mt-tRNAs. Taken together, our results elucidate the molecular mechanism of tRNA 3′ end processing of human mt-tRNA and nu-tRNAs by a single RNase Z enzyme.

Most tRNAs encoded in human mitochondria are highly degenerate and differ notably in sequence, structure and stability from canonical tRNAs encoded in the nucleus[28,69]. Most notably, mt-tRNAs often lack universally conserved tRNA structural features that are used by many tRNA processing factors to recognize nu-tRNAs[23,28,35,45]. ELAC2 catalyzes tRNA 3′ processing both in the nucleus and in the mitochondria[31,47,54] and, thus, must recognize pre-tRNAs from both compartments. Our

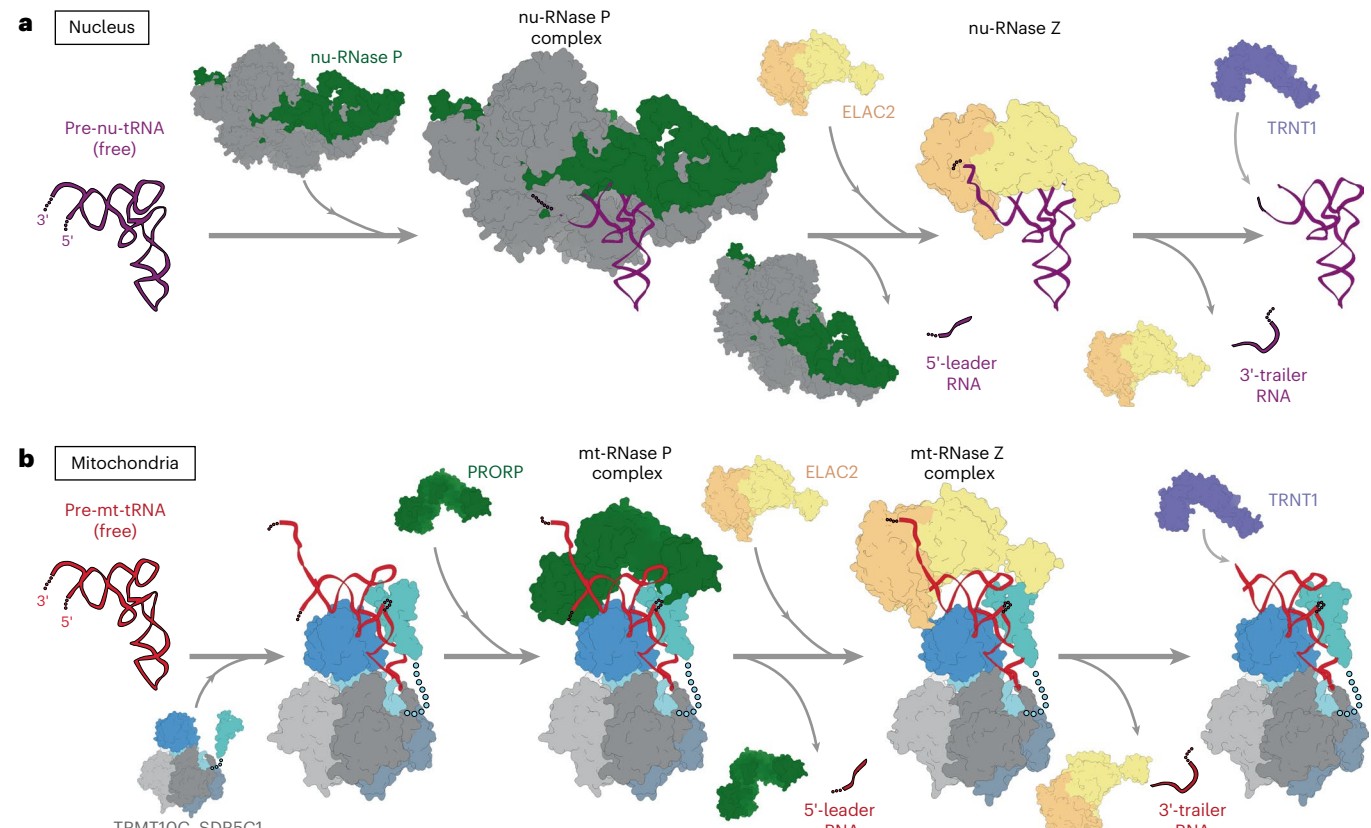

**Fig. 6 | Models of nu-tRNA and mt-tRNA processing. a**, Model of sequential tRNA maturation in the nucleus. Pre-nu-tRNA is shown in purple, the RNA subunit of nu-RNase P is shown in green, the accessory protein subunits are shown in gray, ELAC2 is shown in shades of orange and TRNT1 is shown in light purple. **b**, Model of sequential tRNA maturation in mitochondria. Pre-mt-tRNA is shown in red, TRMT10C is shown in shades of blue, the SDR5C1 tetramer is shown in shades of gray, PRORP is shown in green, ELAC2 is shown in shades of orange and TRNT1 is shown in light purple.

structural and biochemical data reveal that ELAC2 recognizes canonical nu-tRNAs and degenerated mt-tRNAs through distinct mechanisms.

For canonical tRNAs, the elbow structure is recognized by the flexible arm of ELAC2 through interactions with the conserved T-loop and G19–C56 pair. The flexible arm and endonuclease domain of ELAC2 clamp the tRNA core, which positions the 3′ cleavage site near the active site. Hence, disruption of the tRNA elbow would impair the stabilization of the ELAC2 flexible arm and disfavor productive positioning of the 3′ cleavage site. All ELAC2–RNA interactions observed in mt-RNase Z^Gln can be formed in the presence or absence of TRMT10C–SDR5C1. Thus, the ELAC2–RNA interactions observed in mt-RNase Z^Gln are likely the major determinants for recognizing canonical pre-tRNA substrates both in the mitochondria and in the nucleus. A similar mechanism of substrate binding is used by the *Bsu*-RNase Z_S homodimer, suggesting a conserved mechanism of canonical tRNA recognition between long-form and short-form RNase Z enzymes[49].

Interestingly, rather than facilitating 3′ processing of canonical tRNAs by ELAC2, the TRMT10C–SDR5C1 complex appears to slightly inhibit 3′ processing of mt-tRNA^Gln and nu-tRNA^Gly precursors (Fig. 1d). While our data do not provide an obvious mechanistic explanation for this observation, we show that canonical mt-tRNAs nonetheless remain associated with TRMT10C–SDR5C1 following PRORP dissociation and are processed by ELAC2 in complex with TRMT10C–SDR5C1. Thus, it remains unclear whether the slight inhibitory effect of TRMT10C–SDR5C1 on canonical mt-tRNA processing has functional relevance in vivo (for example, in the regulation of mt-tRNA levels).

Most human mt-tRNAs contain reduced or variable D-loops and T-loops that cannot form the canonical elbow structure[28,29,70,71]. Our analysis shows that TRMT10C–SDR5C1 specifically enables 3′ end processing of mt-tRNAs in which the canonical elbow is degenerated. The structure of mt-RNase Z^Tyr shows that TRMT10C–SDR5C1, in addition to its structural support for the overall tRNA fold[45], facilitates processing of such mt-tRNAs by stabilizing ELAC2 through direct protein–protein interactions with TRMT10C, which compensate for the loss of interactions between ELAC2 and the tRNA elbow. Thus, the TRMT10C–SDR5C1 complex appears to have a twofold role in mt-RNase Z. On the one hand, it recognizes and stabilizes the mt-tRNAs in a common L-shaped fold[45]; on the other hand, it acts as a 'prosthetic' extension for degenerated mt-tRNAs lacking stable elbow structures and provides compensatory anchor points for ELAC2. This compensatory role of the TRMT10C–SDR5C1 complex for 3′ processing is analogous to its function in mt-RNase P for 5′ processing, where it similarly stabilizes the mt-tRNA substrate and mediates binding of the PRORP subunit[45,72].

In human mitochondria, mt-tRNA^Ser(AGY) represents the only exception among noncanonical mt-tRNAs that is not efficiently 3′ processed by ELAC2, even in the presence of TRMT10C–SDR5C1 (refs. 46,73). mt-tRNA^Ser(AGY) completely lacks the D-arm and has an extended and substantially remodeled T-arm, thus representing the shortest and most degenerated among all human mt-tRNAs. Consequently, it does not adopt a stable L-shaped fold and lacks interfaces required for stable binding of either TRMT10C or the ELAC2 flexible arm[35]. Thus, cleavage of mt-tRNA^Ser(AGY) is predominantly dependent on processing of the flanking mt-tRNA^His and mt-tRNA^Leu(UUR) and may potentially involve additional protein factors such as YbeY[46,73,74].

The structural data also provide insight into the catalytic mechanism of ELAC2. In the catalytically productive conformation, the phosphodiester group of nucleotide 74 is positioned in the ELAC2 active site, surrounded by the two Zn^2+ ions, the putative general base Asp550 and

residue His702, which was proposed to stabilize the leaving group[66]. The resulting precatalytic configuration of ELAC2 is very similar to that previously proposed for *Bsu*-RNase Z, suggesting a conserved catalytic mechanism among RNase Z enzymes[49,52,66].

Our results also support the previously proposed role of TRMT10C–SDR5C1 as a platform for mt-tRNA maturation[46]. As the structures of mt-RNase Z and mt-RNase P show, TRMT10C requires neither PRORP nor ELAC2 to bind the pre-tRNA and none of the extensive TRMT10C–tRNA interactions are affected by 5′ or 3′ processing[45]. Thus, our results suggest that mt-tRNAs remain bound to the TRMT10C–SDR5C1 complex for multiple maturation steps, consistent with previous biochemical observations, as well as with structures of mt-RNase Z[His] and TRMT10C–SDR5C1–mt-tRNA[Ile]–TRNT1 complexes that were reported during the preparation of this paper[46,75,76]. Additional maturation steps supported by the TRMT10C–SDR5C1 platform following 5′ and 3′ processing may include 3′-CCA addition by TRNT1 and further modifications by modifying enzymes and aminoacyl tRNA synthetases (AARSs). Thus, in conjunction with previous data, our data allow us to propose a model for the sequential maturation of nu-tRNAs and mt-tRNAs (Fig. 6). First, RNase P binds to the pre-tRNA containing the 5′ leader and 3′ trailer to catalyze 5′ processing. ELAC2 is prevented from productive binding to the same substrate at this stage because of steric discrimination against 5′ unprocessed pre-tRNAs. Following 5′ end processing, RNase P must dissociate before ELAC2 can be recruited to the pre-tRNA through interactions of the ELAC2 flexible arm with either the tRNA elbow or the TRMT10C NTD, leading to 3′ processing. The resulting 5′ and 3′ processed tRNA can then be further matured by TRNT1, which catalyzes the addition of the 3′-CCA end. In the nucleus, the enzymes carrying out these maturation steps are self-sufficient, with nuclear RNase Z acting as a single-subunit enzyme (Fig. 6a). By contrast, in mitochondria, all three maturation steps take place on pre-tRNAs stably associated with TRMT10C–SDR5C1 (Fig. 6b).

These observations lead us to hypothesize that the TRMT10C–SDR5C1 complex may have evolved as a common solution for multiple mt-tRNA-binding factors to the common challenge of substrate recognition posed by the structural degeneration of mt-tRNAs[33,35]. The canonical tRNA elbow structure has a key role in cytosolic gene expression machineries, as it is recognized by many tRNA-binding factors including RNase P enzymes, RNase Z enzymes, CCA-nucleotidyltransferases, AARSs and the ribosome[23,42,52,77–79]. Its loss in mt-tRNAs, thus, imposes strong pressure to evolve compensatory mechanisms to maintain these functional interactions. In human mt-RNase P and mt-RNase Z, the PRORP pentatricopeptide repeat domain and ELAC2 flexible arm, both of which classically interact with the tRNA elbow, interact with the TRMT10C NTD through the same surface on its globular subdomain. Notably, this globular subdomain first appears and becomes fixed in TRMT10 homologs of bilaterians, coinciding with the widespread erosion of mt-tRNAs[30] (Extended Data Fig. 10). Thus, the TRMT10C–SDR5C1 complex may have evolved to specifically compensate for the structural erosion of mt-tRNAs and to maintain their indispensable function in mitochondrial gene expression. This is further supported by observations from nematodes, which possess some of the most reduced mt-tRNAs in the animal lineage. Nematode TRMT10C and ELAC2 appear to have acquired additional features not found in any other bilaterian animals, accompanying the evolution of specialized mitochondrial translation elongation factors (mt-EF-Tu1 and mt-EF-Tu2) for binding of highly eroded D-armless and T-armless mt-tRNAs[80,81]. Together, these observations highlight the general evolutionary trend that nucleus-encoded proteins evolve to compensate for the ever-increasing mutational load in mt-RNAs[81–83].

In summary, our results explain how ELAC2 recognizes and catalyzes 3′ end processing of two structurally divergent sets of tRNAs found in human mitochondria and the nucleus. Together with previous observations, these results further highlight the role of the conserved tRNA elbow structure in tRNA recognition and provide insight into the evolutionary history of TRMT10C–SDR5C1 as a platform for mt-tRNA maturation.

## Online content

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

## Methods

No statistical methods were used to predetermine sample size. The experiments were not randomized and the investigators were not blinded to allocation during experiments and outcome assessment.

### Protein cloning, expression and purification

PRORP, TRMT10C and SDR5C1 were cloned, expressed and purified as previously reported[45]. Mutant variants of TRMT10C were generated by restriction-free cloning PCR using the vector encoding wild-type TRMT10C and SDR5C1 as the template (Supplementary Data 1). TRMT10C–SDR5C1 complexes containing mutant TRMT10C were expressed and purified following the same protocol as for the wild-type proteins. The sequence encoding ELAC2 lacking the N-terminal mitochondrial targeting sequence (Δ1–31) was PCR-amplified from human complementary DNA (cDNA) obtained from B lymphocytes (kind gift from P. Cramer) (Supplementary Table 1). It was cloned into the 438-B vector (kind gift from S. Gradia; Addgene plasmid 55219; RRID:Addgene_55219)[85] in frame with an N-terminal 6xHis-tag followed by a tobacco etch virus (TEV) cleavage site, followed by the generation of baculovirus. ELAC2 mutants were generated by site-directed mutagenesis using round-the-horn PCR amplification of wild-type ELAC2 in the 438-B vector (Supplementary Data 1). Baculoviruses for ELAC2 and its variants were generated using Sf9 (Oxford Expression Technologies, 600100) and Sf21 (Expresson Systems, 94-003F) cells (kind gift from P. Cramer) and were expressed in an insect cell expression system using Hi-Five cells (Expression Systems, 94-002F; kind gift from P. Cramer) as previously described[86,87]. All subsequent steps were carried out at 4 °C. Cells were harvested by centrifugation at 238$g$ for 30 min and resuspended in lysis buffer at pH 7.5 containing 20 mM Na–HEPES, 300 mM NaCl, 40 mM imidazole, 1 mM DTT, 10% glycerol and 1× protease inhibitor cocktail (Roche) or 0.284 μg ml$^{-1}$ leupeptin, 1.37 μg ml$^{-1}$ pepstatin, 0.17 mg ml$^{-1}$ phenylmethylsulfonyl fluoride and 0.33 mg ml$^{-1}$ benzamidine. Cells were lysed by sonication, followed by sequential centrifugation in an A27 rotor (Thermo Fisher Scientific) at 26,195$g$ for 30 min and ultracentrifugation in a Type 45 Ti rotor (Beckman Coulter) for 60 min. The supernatant was filtered through filtration membranes with a sieve width of 5 μm followed by 0.8 μm and applied to a HisTrap HP 5-ml column (Cytiva) equilibrated with lysis buffer. The column was washed with 10 column volumes (CV) of lysis buffer, followed by 10 CV of high-salt wash buffer (50 mM Na–HEPES, 1 M NaCl, 40 mM imidazole, 10% glycerol and 2 mM DTT, pH 7.5), followed again by 5 CV of lysis buffer. Bound proteins were eluted with 9.5 CV of elution buffer (17 mM Na–HEPES, 255 mM NaCl, 334 mM imidazole, 8.5% glycerol and 1.7 mM DTT, pH 7.5). To the eluted protein, 1.5 mg of recombinant TEV protease (homemade) was added, followed by overnight dialysis in 50 mM Na–HEPES, 300 mM NaCl, 10% glycerol and 2 mM DTT at pH 7.5. Dialyzed eluate was reapplied to the HisTrap HP column equilibrated with buffer containing 50 mM Na–HEPES, 300 mM NaCl, 10 mM imidazole, 10% glycerol and 2 mM DTT at pH 7.5 and washed with 10 CV of the same buffer. The flowthrough and wash were collected and applied to a HiTrap Heparin 5-ml column (Cytiva) equilibrated with 50 mM Na–HEPES, 150 mM NaCl, 10% glycerol and 2 mM DTT at pH 7.5. Bound proteins were eluted with a gradient of 150–1,000 mM NaCl with 50 mM Na–HEPES, 10% glycerol and 2 mM DTT at pH 7.5. Fractions containing ELAC2 according to SDS–PAGE were pooled and further purified using a Superdex 200 Increase 10/300 GL column (Cytiva) equilibrated with 20 mM Na–HEPES, 150 mM NaCl, 10% glycerol and 5 mM DTT at pH 7.5. Fractions containing ELAC2 were concentrated using Amicon Ultra-4 30-kDa centrifugal filter devices (Merck Millipore), aliquoted, flash-frozen in liquid nitrogen and stored at −70 °C.

### Preparation of substrate RNAs

Sequences encoding the pre-tRNA substrates under the control of a T7 RNA polymerase promoter were either purchased as gBlocks

(Integrated DNA Technologies) or cloned into a pUC19 vector (Supplementary Table 1). Mutations were introduced by site-directed mutagenesis PCR. The initial templates were amplified by PCR using forward and reverse primers complementary to T7 promoter and the 3′ trailer sequence, respectively (Supplementary Table 1). PCR-amplified templates were purified using QIAquick or MinElute PCR purification kits (Qiagen) and used for run off in vitro transcription (IVT).

The RNA substrates for structural studies (tRNA$^{Tyr}$ with no 5′ leader and 24-nt-long 3′ trailer (0–tRNA$^{Tyr}$–24) and tRNA$^{Gln}$ with 5-nt-long 5′ leader and same 24-nt-long 3′ trailer as tRNA$^{Tyr}$ (5–tRNA$^{Gln}$–24)) were transcribed in reactions containing 1× T7 RNA polymerase reaction buffer (New England Biolabs), 0.001% (w/v) Triton-X 100, 30 mM MgCl$_2$, 4 mM nucleoside triphosphates (NTPs), 5 U per μl T7 RNA polymerase (New England Biolabs) and 0.2 μg μl$^{-1}$ template DNA. The Mg$_2$P$_2$O$_7$ precipitate formed during IVT was solubilized with EDTA at a final concentration of 37 mM and removed by centrifugation. RNAs were then purified by anion-exchange chromatography using a RESOURCE Q 6-ml column (Cytiva). The column was equilibrated with 9.5 CV of buffer A containing 50 mM NaCH$_3$COO and 2 mM MgCl$_2$ at pH 5.5, followed by application of the IVT reaction. Bound RNAs were eluted with a linear gradient from 0% to 100% buffer B containing 1000 mM NaCl, 50 mM NaCH$_3$COO and 2 mM MgCl$_2$ at pH 5.5. Elution fractions were analyzed by urea PAGE, fractions of interest were pooled and RNA was precipitated with NaCH$_3$COO and ethanol at final concentrations of 300 mM and 70%, respectively. Pure RNAs were dissolved in nuclease-free water and stored at −20 °C until further use.

All tRNA substrates used for biochemical analysis were transcribed in reactions containing 1× T7 RNA polymerase reaction buffer (Thermo Fisher Scientific), 2 mM DTT, 30 mM MgCl$_2$, 6 mM of each NTP, 0.002 U per μl *Escherichia coli* PPIase (New England Biolabs), 5 U per μl T7 RNA polymerase (Thermo Fisher Scientific) and 50 ng μl$^{-1}$ template DNA. For internally FAM-labeled tRNAs used in FA experiments, the reactions were spiked with an additional 5% FAM-UTP (Jena Bioscience). After incubation at 37 °C for 10–14 h, reactions were stopped by addition of an equal volume of 2× TBE–urea sample buffer (Thermo Fisher Scientific) and boiling at 95 °C for 5 min. Samples were then separated by 10% urea PAGE and visualized by ultraviolet shadowing; bands containing the RNA of interest were cut and transferred to an RNase-free Eppendorf tube. Gel slices were crushed and RNAs were extracted in buffer containing 300 mM NaCH$_3$COO, 1 mM EDTA and 20 mM Tris at pH 5. Extracted RNAs were ethanol-precipitated and stored as described above.

### RNA cleavage assays

Pre-tRNA cleavage assays were carried out in buffer containing 20 mM HEPES–KOH pH 7.4, 150 mM KCl, 3 mM MgCl$_2$, 10 μM ZnCl$_2$, 2 mM DTT and 100 μM S-adenosyl L-methionine (SAM). Reactions were set up by adding 200 nM pre-tRNA to the reaction buffer, followed by incubation for 10 min at 30 °C in the presence or absence of 800 nM TRMT10C–SDR5C1 complex. Reactions were started by the addition of 50 nM ELAC2, followed by incubation at 30 °C for 20 min, unless stated otherwise. For control reactions on 5′ leader-containing pre-tRNA substrates, 50 nM PRORP was added together with ELAC2. Reactions were stopped by the addition of 2× TBE–urea sample buffer (Thermo Fisher Scientific) supplemented with proteinase K (Thermo Fisher Scientific) and incubated at 50 °C for 30 min. Samples were boiled for 5 min at 95 °C and then loaded onto 15% TBE–urea PAGE. Gels were subsequently soaked in TBE buffer containing 1× SYBR gold nucleic acid gel stain (Thermo Fisher Scientific) and visualized on a Typhoon imager (Cytiva).

### FA

FAM-labeled pre-mt-tRNAs (20 nM) with no 5′ leader and 24-nt-long 3′ trailer were incubated with serial dilutions of purified TRMT10C–SDR5C1 complex or ELAC2-Asp550Asn (starting from 8.3 μM) in FA buffer (20 mM Tris-HCl pH 8, 80 mM NaCl, 40 mM KCl, 3 mM MgCl$_2$,

5% glycerol and 2 mM DTT) at 20 °C for 20 min. After incubation, 20 µl of each binding reaction was transferred into a black flat-bottom 384-well microplate (Greiner) and FA measurements were performed on a Sparc Plate Reader (Tecan) using SPARKCONTROL version 3.1 with 485-nm excitation and 535-nm emission wavelengths (each with a bandwidth of 20 nm) at room temperature. Each experiment was performed in triplicate and the obtained data were analyzed using Prism version 10.2.3 (GraphPad) software. Binding curves were fit with a single-site quadratic binding equation:

$$y = \frac{B_{max} \times \left([P] + [R] + K_d - \sqrt{([P] + [R] + K_d)^2 - 4([P] \times [R])}\right)}{2 \times [R]}$$

where $B_{max}$ is the maximum specific binding, $[R]$ is the concentration of tRNA, $[P]$ is the concentration of TRMT10C–SDR5C1 complex or ELAC2 and $K_d$ is the apparent disassociation constant for protein–tRNA complexes.

## Cryo-EM sample preparation and data collection

Δ1–91 TRMT10C was used for the structural studies to avoid excessive particle clustering observed with full-length TRMT10C. For assembly of mt-RNase Z$^{Gln}$ complex, 2.7 nmol of 5–tRNA$^{Gln}$–24 substrate was first subjected to 5′ processing with 2.7 nmol of Δ1–91 TRMT10C–SDR5C1 complex and 270 pmol of Δ1–45 PRORP in a reaction buffer at pH 8.0 containing 25 mM Tris-HCl, 150 mM NaCl, 5 mM MgCl$_2$, 5 mM DTT and 200 µM SAM and incubated at 30 °C for 16 h. The reaction buffer was then exchanged to a buffer at pH 8.0 containing 25 mM Tris-HCl, 20 mM NaCl, 20 µM ZnCl$_2$, 40 µM SAM and 4 mM DTT using Zeba spin desalting columns (Thermo Fisher Scientific). Then, 7.0 nmol of Δ1–31 ELAC2-Asp550Asn was added and the reaction incubated on ice for 60 min. For assembly of mt-RNase Z$^{Tyr}$ complex, 2.6 nmol of 0–tRNA$^{Tyr}$–24 substrate was mixed with 1.1 nmol of Δ1–91 TRMT10C–SDR5C1 and 3 nmol of ELAC2-Asp550Asn in buffer at pH 8.0 containing 25 mM Tris-HCl, 25 mM NaCl, 2 mM DTT, 10 µM ZnCl$_2$ and 20 µM S-adenosyl homocysteine (SAH) and incubated on ice for 60 min at 4 °C. Both assembled complexes were then used for GraFix[88] in a 10–30% sucrose density gradient at pH 7.5 containing 25 mM Na–HEPES, 20 mM NaCl, 20 µM ZnCl$_2$ and 2 mM DTT with and without a glutaraldehyde gradient (0–0.015% for mt-RNase Z$^{Gln}$ and 0–0.025 % for mt-RNase Z$^{Tyr}$). Ultracentrifugation was carried out in a SW 60 Ti swinging-bucket rotor (Beckman Coulter) at 40,000 rpm at 10 °C for 16 h. The gradient solutions were divided into 200-µl fractions and analyzed by SDS–PAGE and a Nanodrop One spectrophotometer (Thermo Fisher Scientific). Fractions 14–17 from the gradient with glutaraldehyde crosslinker were pooled for both complexes for cryo-EM on the basis of SDS–PAGE profiles for both samples without the crosslinker (Extended Data Fig. 2c–f). Pooled fractions were buffer-exchanged to 25 mM Na–HEPES, 20 mM NaCl, 20 µM ZnCl$_2$ and 2 mM DTT at pH 7.5 and concentrated using Amicon Ultra 0.5-ml 10-kDa-cutoff centrifugation devices (Merck Millipore). Then, 4 µl of the concentrated sample was applied to freshly glow-discharged R2/1 holey carbon grids (Quantifoil), blotted with blot force of 5 for 5 s using a Vitrobot Mark IV (Thermo Fisher Scientific) at 95% humidity and 4 °C and plunge-frozen in liquid ethane.

Cryo-EM data collection was performed with SerialEM[89] using a Titan Krios transmission electron microscope (Thermo Fisher Scientific) operated at 300 keV. Images were acquired in energy-filtered transmission EM mode using a GIF quantum energy filter set to a slit width of 20 eV and a K3 direct electron detector (Gatan) at a nominal magnification of ×105,000, corresponding to a calibrated pixel size of 0.834 Å per pixel. Exposures were saved as nonsuper-resolution counting image stacks of 40 video frames, with electron doses of 0.99–1.05 e$^-$ per Å$^2$ per frame. Image stacks were acquired with stage movement per hole for the mt-RNase Z$^{Tyr}$ dataset and with stage movement per

3 × 3 holes with active beam-tilt compensation for the mt-RNase Z$^{Gln}$ dataset, as implemented in SerialEM.

## Cryo-EM data processing and analysis

Image stacks were preprocessed on the fly with gain correction, motion correction, contrast transfer function (CTF) estimation, particle picking and extraction at 2.5 Å per pixel using Warp version 1.0.9 (ref. 90).

For mt-RNase Z$^{Gln}$ dataset, 12,168,828 particles autopicked from 39,387 micrographs were subjected to two-dimensional (2D) classification in cryoSPARC (Extended Data Fig. 3a). Particles belonging to classes clearly lacking protein-like features were discarded, while the 11,090,926 particles belonging to remaining classes were divided into 'good' or 'bad' subsets. The bad particle subset was used for ab initio reconstruction resulting in five bad references. A single good reference was obtained from a previous smaller dataset of mt-RNase Z$^{Gln}$ complex processed in cryoSPARC version 4.2.1 (Structura Biotechnology)[91]. All 11,090,926 particles were subjected to supervised 3D classification (heterogeneous refinement algorithm in cryoSPARC) using the good and bad initial references. A total of 4,192,922 particles belonging to the good class were subjected to another 2D classification in cryoSPARC, from which 3,255,655 particles were selected. These particles were further subjected to unsupervised 3D classification in RELION version 3.1.0 (ref. 92), which yielded in a single class of 509,418 particles leading to an isotropic reconstruction of the TRMT10C–SDR5C1–pre-tRNA$^{Gln}$ module. These particles were re-extracted at 0.834 Å per pixel and used for a consensus refinement and subsequent CTF refinement in RELION. To classify with respect to heterogeneity around the ELAC2-binding site, focused 3D classification without image alignment was carried out in cryoSPARC with a mask around the ELAC2-binding site, resulting in 152,762 particles with robust ELAC2 density. After consensus refinement and a focused refinement in cryoSPARC centered on the ELAC2 density, another focused 3D classification was carried out in cryoSPARC with mask around the 3′ trailer RNA and ELAC2 active site, resulting in a class of 75,812 particles with ordered 3′ trailer density. These particles were subjected to a consensus 3D refinement and focused refinement in cryoSPARC centered of the ELAC2 density. The resulting consensus and ELAC2-focused maps were resharpened with a B factor of 0 Å$^2$ to avoid oversharpening of the flexible regions and combined using PHENIX[93], resulting in a composite map of the mt-RNase Z$^{Gln}$ complex[77].

For mt-RNase Z$^{Tyr}$ dataset, 5,987,454 particles autopicked from 15,344 micrographs were subjected to 2D classification, from which particles clearly lacking RNase Z-like features were discarded (Extended Data Fig. 3b). The remaining 5,275,530 particles were divided into good and bad subsets and the particles from the bad subset were used to generate four bad initial references using the ab initio reconstruction algorithm in cryoSPARC. A low-resolution consensus map of mt-RNase Z$^{Tyr}$ complex from a previous smaller dataset was used as the good initial reference. All 5,275,530 particles were subjected to a supervised 3D classification in cryoSPARC with the four bad and a single good initial reference. The resulting 1,780,577 particles belonging to the good class were subjected to a focused 3D classification with respect to the density around the ELAC2-binding site in RELION. A total of 482,446 particles containing robust density at the ELAC2-binding site were re-extracted at 0.834 Å per pixel and subjected to an unsupervised 3D classification in RELION, which yielded a single class of 227,594 particles resulting in an isotropic reconstruction of the complex. These particles were further cleaned up using 2D classification in cryoSPARC and a second focused 3D classification around ELAC2 density, resulting in a class of 57,585 particles, which resulted in the high-resolution map of ELAC2 at 3.2 Å. These particles were further classified with respect to the density around the ELAC2 flexible arm, resulting in a class with 28,023 particles showing ordered ELAC2 flexible arm density. These particles were subjected to a consensus refinement and a focused 3D refinement around ELAC2 density using cryoSPARC. The resulting maps were resharpened with a B factor of 0 Å$^2$ to avoid oversharpening the

flexible regions and combined in PHENIX[93], resulting in the composite map of the mt-RNase Z[Tyr] complex.

The 3D variability analysis for both complexes was performed in cryoSPARC[91] using alignments and reference volumes from the consensus refinement, solving for two principal modes of covariance. To resolve the productive active-site conformation in mt-RNase Z[Tyr], outputs of the consensus refinement of the 57,585 particles resulting in the high-resolution map of ELAC2 were used. Particles were binned into three subsets on the basis of their reaction coordinates along the first principal component. The subsets were independently refined with a refinement mask around the ELAC2 density, resulting in density maps for the three subsets, showing the 'extended', 'relaxed' and 'contracted' conformation of the pre-tRNA. Local resolution estimations were calculated in cryoSPARC[91]. Angular distribution plots were generated using a script packaged with Warp version 1.0.9 (ref. 90).

#### Model building and refinement
The initial models for TRMT10C, SDR5C1 and pre-tRNA[Tyr] were obtained from the model of mt-RNase P[Tyr] complex (Protein Data Bank (PDB) 7ONU)[45] and the initial model for ELAC2 was obtained from Alpha-FoldDB (Q9BQ52). The models were rigid-body fitted into the final maps of mt-RNase Z[Gln] or mt-RNase Z[Tyr] complex using UCSF ChimeraX 1.6.1 and rebuilt and refined using WinCoot version 0.9.8.7 (ref. 94). For mt-RNase Z[Gln], the residues of pre-tRNA[Tyr] were iteratively substituted to match the sequence of pre-tRNA[Gln]. The complete models of mt-RNase Z[Gln] and mt-RNase Z[Tyr] complexes were refined against the respective composite maps using ISOLDE[95] and phenix.real_space_refine[93,96], followed by a single round of ADP refinement in PHENIX[93,96] against the respective consensus maps. The MolProbity package within PHENIX suite was used for model validation[97]. The trajectory of the RNA in the productive substrate-engaged ELAC2 active site was based on the contracted conformation of pre-mt-tRNA[Tyr] (Extended Data Fig. 8d,e)[79]. All structural analyses and image renderings for figure preparation were performed using UCSF ChimeraX 1.6.1 (refs. 98,99) or PyMol 2.0 (Schrödinger).

#### Reporting summary
Further information on research design is available in the Nature Portfolio Reporting Summary linked to this article.

#### Data availability
The UniProt accession numbers for ELAC2, TRMT10C and SDR5C1 are Q9BQ52, Q7L0Y3 and Q99714. The structure coordinates for mt-RNase P[Tyr], GlyRS-bound nu-tRNA[Gly], yeast tRNA[Phe] and human nu-RNase P were obtained from the PDB under accession codes 7ONU, 5E6M, 4TNA and 6AHU. The cryo-EM density reconstructions for mt-RNase Z[Gln] and mt-RNase Z[Tyr] were deposited to the EM Database under accession codes EMD-19455 and EMD-19457. The respective structure coordinates were deposited to the PDB under accession codes 8RR3 and 8RR4. Source data are provided with this paper.

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

### Acknowledgements
We thank all members of the Hillen lab for help and discussions. We thank C. Dienemann and U. Steuerwald (Max Planck Institute for Multidisciplinary Sciences cryo-EM facility) for assistance with cryo-EM data acquisition and P. Prado for help with formatting the paper. We thank S. Gradia (University of California, Berkeley) for the plasmid expression vectors and P. Cramer (Max Planck Institute for Multidisciplinary Sciences Göttingen) for insect cell expression cell lines and human cDNA. This work was funded by the Deutsche Forschungsgemeinschaft (DFG) under Germany's Excellence Strategy EXC 2067/1-390729940 (to H.S.H), FOR2848 (to H.S.H.), SFB1190 (to H.S.H.) and SFB1565 (project number 469281184 to H.S.H. and K.E.B.) and by the European Union (European Research Council starting grant MitoRNA, grant no. 101116869 to H.S.H.). Views and opinions expressed are those of the authors only and do not necessarily reflect those of the European Union or the European Research Council Executive Agency. Neither the European Union nor the granting authority can be held responsible for them. H.S.H. was supported by the European Molecular Biology Organization Young Investigator program. This work was supported by the 'Molecular Biology' Ph.D. program at the International Max Planck Research School (University of Göttingen). A.B. and R.D.Y were doctoral students of the 'Molecular Biology' Ph.D. program at the International Max Planck Research School and the Göttingen Graduate School for Neurosciences, Biophysics and Molecular Biosciences (DFG grant GSC 226) at Georg-August University of Göttingen. The funders had no role in study design, data collection and analysis, decision to publish or preparation of the paper.

### Author contributions
A.B. purified proteins, reconstituted and prepared cryo-EM grids of mt-RNase Z[Tyr], collected and processed cryo-EM data and built and analyzed both structural models, B.K. purified proteins and RNAs and carried out biochemical assays, R.D.Y. purified proteins, reconstituted and prepared cryo-EM grids of mt-RNase Z[Gln] and collected cryo-EM

data, and L.S. purified proteins and prepared cryo-EM grids of mt-RNase Z^Tyr together with A.B. K.D. performed insect cell culture and protein expression. H.S.H. conceptualized the study and supervised the research. A.B., B.K., K.E.B. and H.S.H. interpreted data and wrote the paper.

## Funding

## Competing interests

The authors declare no competing interests.

## Additional information

**Extended data** is available for this paper at https://doi.org/10.1038/s41594-024-01445-w.

**Correspondence and requests for materials** should be addressed to Hauke S. Hillen.

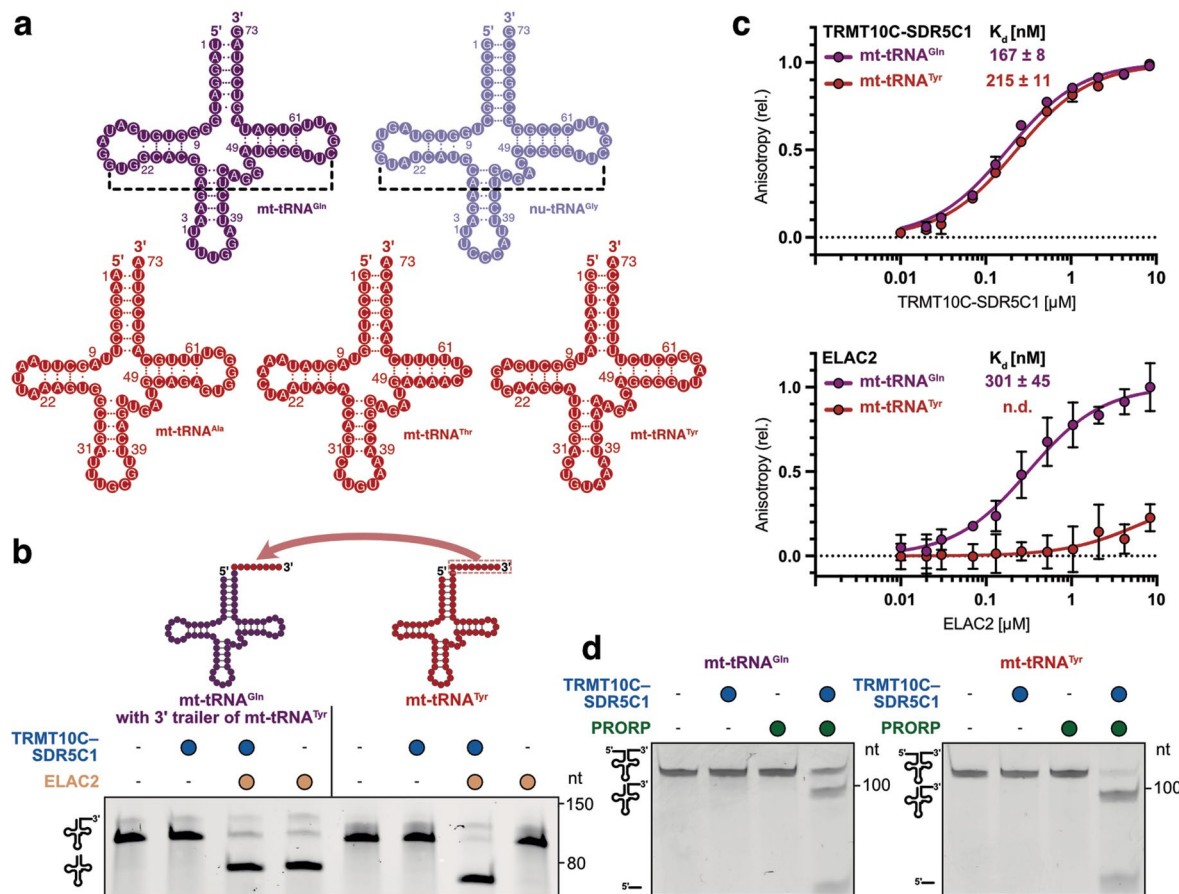

**Extended Data Fig. 1 | Differential dependence of human tRNAs on TRMT10C−SDR5C1 for 3′-processing. a**. Schematic cloverleaf representation of all human tRNAs used in this study. Mt-tRNAs lacking the canonical elbow structures and requiring TRMT10C−SDR5C1 for 3′-processing: mt-tRNA[Ala], mt-tRNA[Thr] and mt-tRNA[Tyr], are shown in red; mt-tRNA[Gln] and nu-tRNA[Gly], which possess canonical elbow interactions and do not require TRMT10C−SDR5C1 for 3′-processing, are shown in dark purple and light purple, respectively. G-C, A-U and G-U base pairs are indicated by three, two and one dots, respectively. Canonical G19-C56 base pairs are indicated by dashed black line segments. **b**. *In vitro* cleavage assay showing that the differential dependence of mt-tRNA[Gln] and mt-tRNA[Tyr] on TRMT10C−SDR5C1, as observed in Fig. 1d, does not relate to differences in the 3′-trailer sequences. The mt-tRNA[Gln] construct used in this

assay contains the 3′-trailer sequence derived from mt-tRNA[Tyr]. Representative gel of three independent replicates. **c**. Fluorescence anisotropy of FAM-labeled mt-tRNA[Gln] and mt-tRNA[Tyr] precursors with TRMT10C−SDR5C1 complex (top) and ELAC2(D550N) (bottom). Each dot represents the mean from three replicates with standard deviations shown as error bars. Protein-RNA dissociation constants ($K_d$) were determined by fitting the data with a single site quadratic binding equation (Methods). The binding data did not provide a reliable estimate for the $K_d$ between mt-tRNA[Tyr] and ELAC2(Asp550Asn). **d**. *In vitro* 5′ cleavage assay with PRORP, showing its dependence on TRMT10C−SDR5C1 for processing of both mt-tRNA[Gln] and mt-tRNA[Tyr]. Representative gels of three independent replicates.

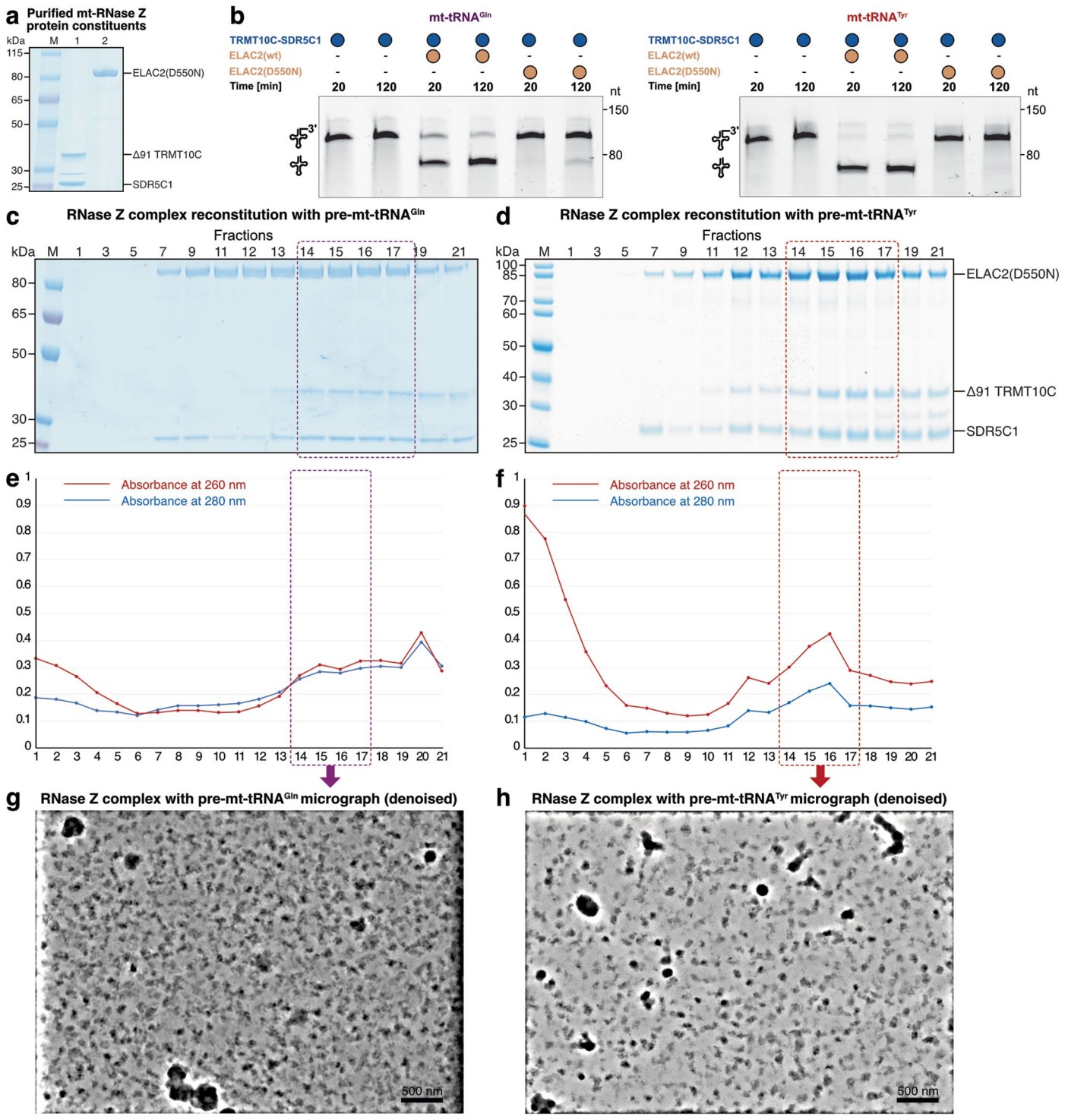

**Extended Data Fig. 2 | *In vitro* reconstitution of mt-RNase Z complexes.**
**a**. SDS-PAGE showing purified protein components of the mt-RNase Z complex. **b**. *In vitro* cleavage assay showing the reduction in catalytic activity of the ELAC2(Asp550Asn) variant used for complex reconstitution. **c**. SDS-PAGE analysis of the fractions of a representative sucrose density gradient ultracentrifugation (SDG) without glutaraldehylde crosslinker for mt-RNase Z^Gln. The sample used for cryo-EM was prepared in the presence of the crosslinker. **d**. SDS-PAGE analysis of the fractions of a representative sucrose density gradient ultracentrifugation (SDG) without glutaraldehylde crosslinker for mt-RNase Z^Tyr. The sample used for cryo-EM was prepared in the presence of the crosslinker. **e**. Line graph showing absorbance at 260 nm and 280 nm for SDG fractions for mt-RNase Z^Gln. **f**. Line graph showing absorbance at 260 nm and 280 nm for SDG fractions for mt-RNase Z^Tyr. **g**. A denoised representative cryo-EM micrograph of mt-RNase Z^Gln from a total of 39,387 collected micrographs. **h**. A denoised representative cryo-EM micrograph of mt-RNase Z^Tyr from a total of 15,344 collected micrographs.

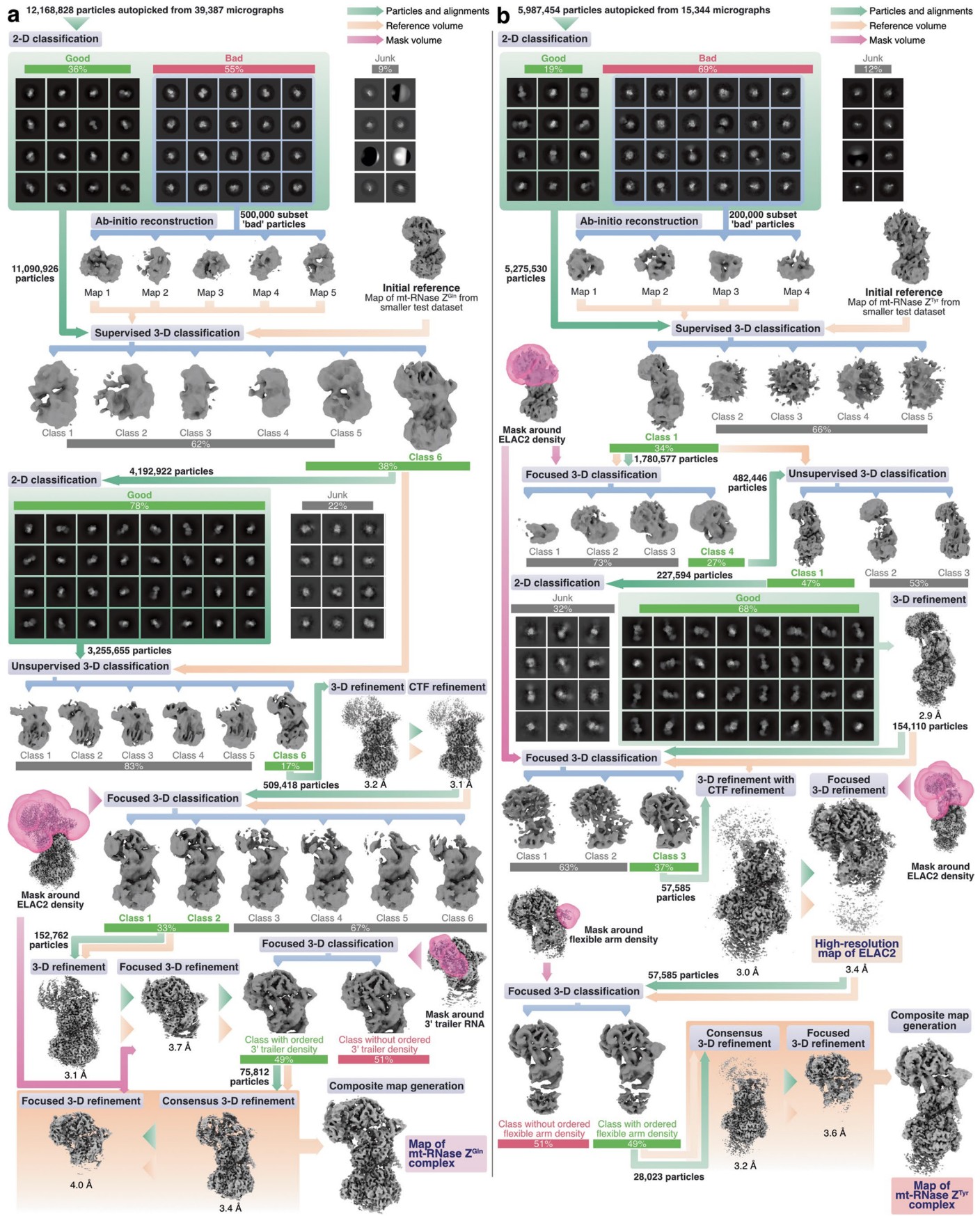

**Extended Data Fig. 3 | Cryo-EM processing. a.** Cryo-EM processing workflow for mt-RNase Z$^{Gln}$ complex. **b.** Cryo-EM processing workflow for mt-RNase Z$^{Tyr}$ complex.

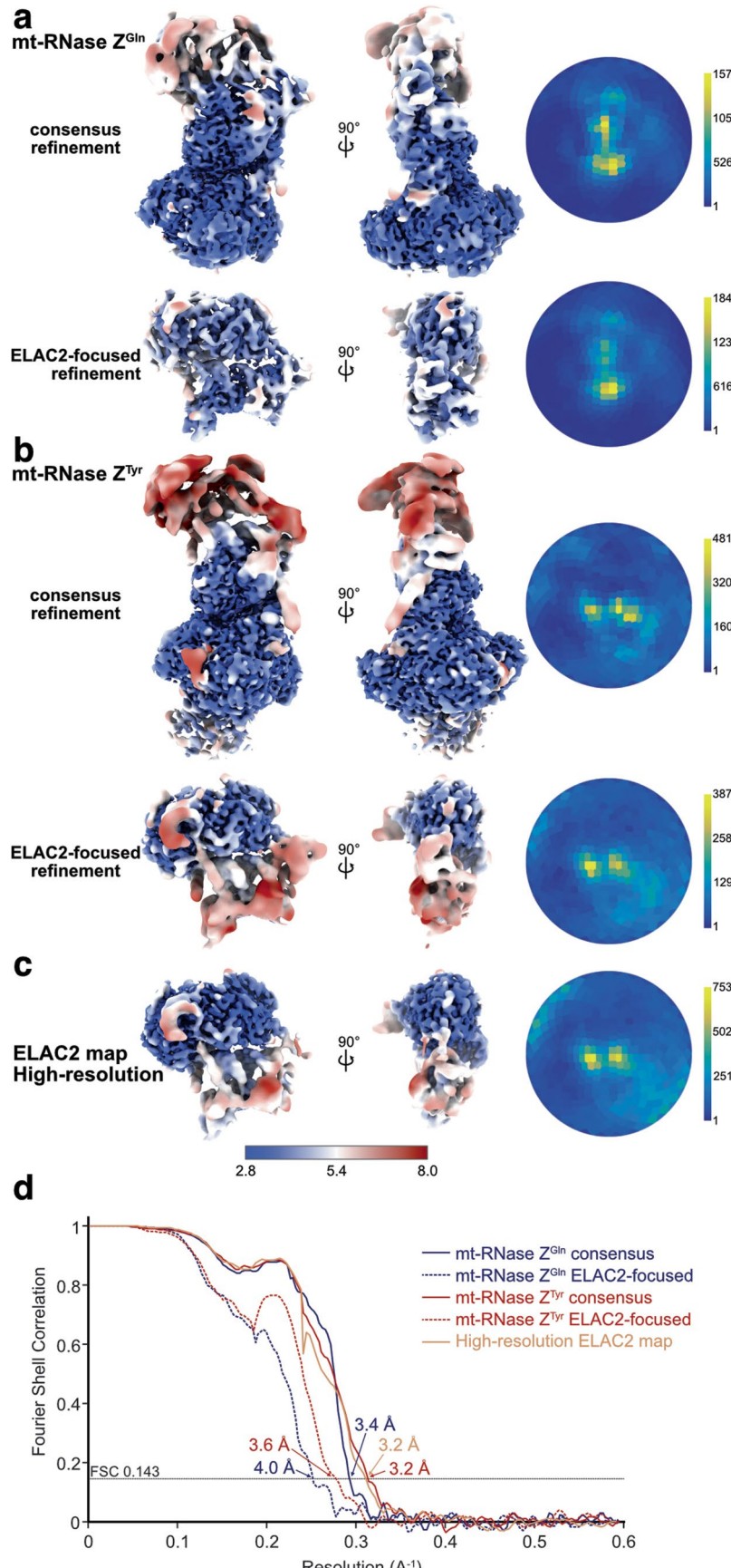

**Extended Data Fig. 4 | Cryo-EM post-processing analysis. a**. Local resolution maps (left) and angular distribution plots (right) for consensus refinement map and ELAC2-focused map of mt-RNase Z^Gln. **b**. Local resolution maps (left) and angular distribution plots (right) for consensus refinement map and ELAC2-focused map of mt-RNase Z^Tyr. **c**. Local resolution map (left) and angular distribution plot (right) for high-resolution ELAC2-focused map from the mt-RNase Z^Tyr dataset. **d**. Fourier Shell Correlation (FSC) plots for maps in a-d. FSC threshold of 0.143 was used for reporting resolutions.

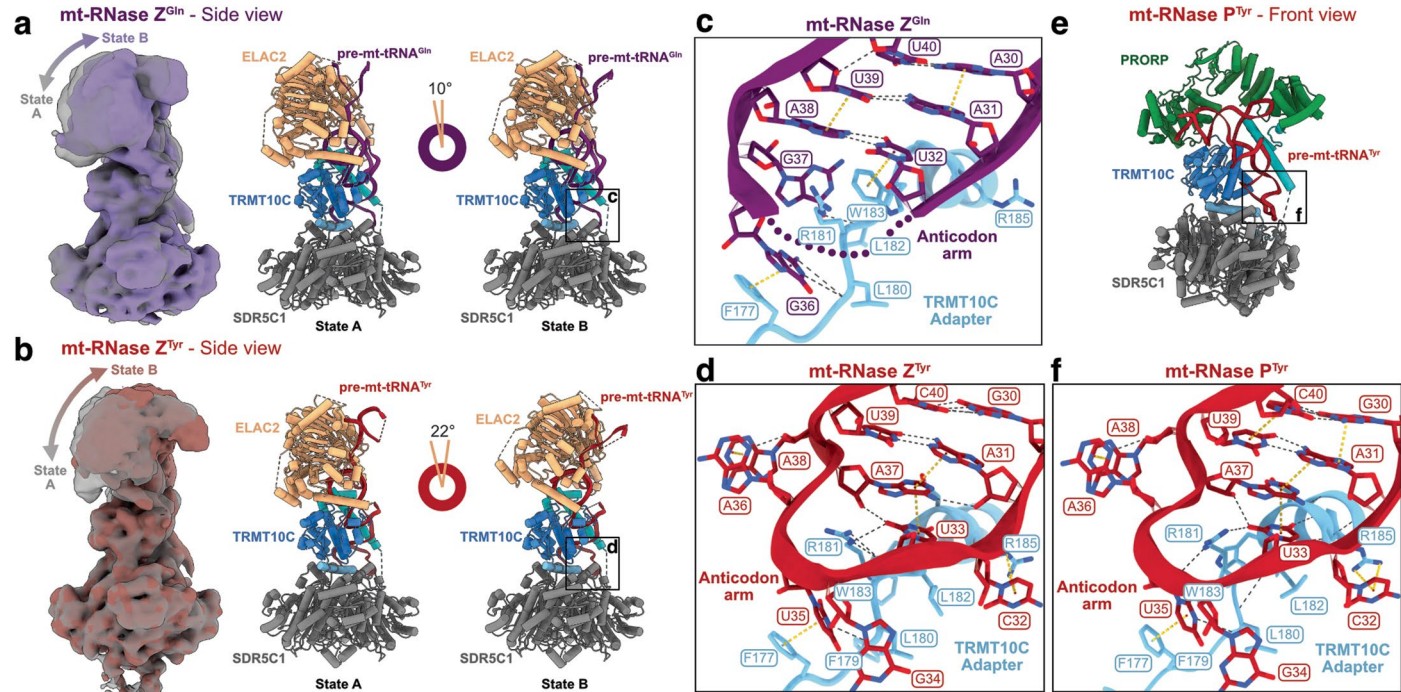

**Extended Data Fig. 5 | Structural comparison of mt-RNase Z^Gln, mt-RNase Z^Tyr and mt-RNase P^Tyr complexes. a**. Conformational variability of ELAC2 in the mt-RNase Z^Gln complex. States A and B represent two ends of a continuum of variability. **b**. Conformational variability of ELAC2 in the mt-RNase Z^Tyr complex. ELAC2 exhibits higher conformational variability in mt-RNase Z^Tyr compared to mt-RNase Z^Gln. **c**. Interactions between TRMT10C and the mt-tRNA^Gln anticodon stem-loop in mt-RNase Z^Gln. The A38-U32 base pair extends the anticodon stem by

one base pair and leads to a distinct stem-loop topology compared to mt-tRNA^Tyr (**d,f**). **d**. Interactions between TRMT10C and mt-tRNA^Tyr anticodon stem-loop mt-RNase Z^Tyr. The anticodon stem-loop topology is similar to that previously observed in mt-RNase P^Tyr (**e,f**)[45]. **e**. Structural model of mt-RNase P^Tyr (PDB: 7ONU)[45]. **f**. Interactions between TRMT10C and the mt-tRNA^Tyr anticodon stem-loop in mt-RNase P^Tyr.

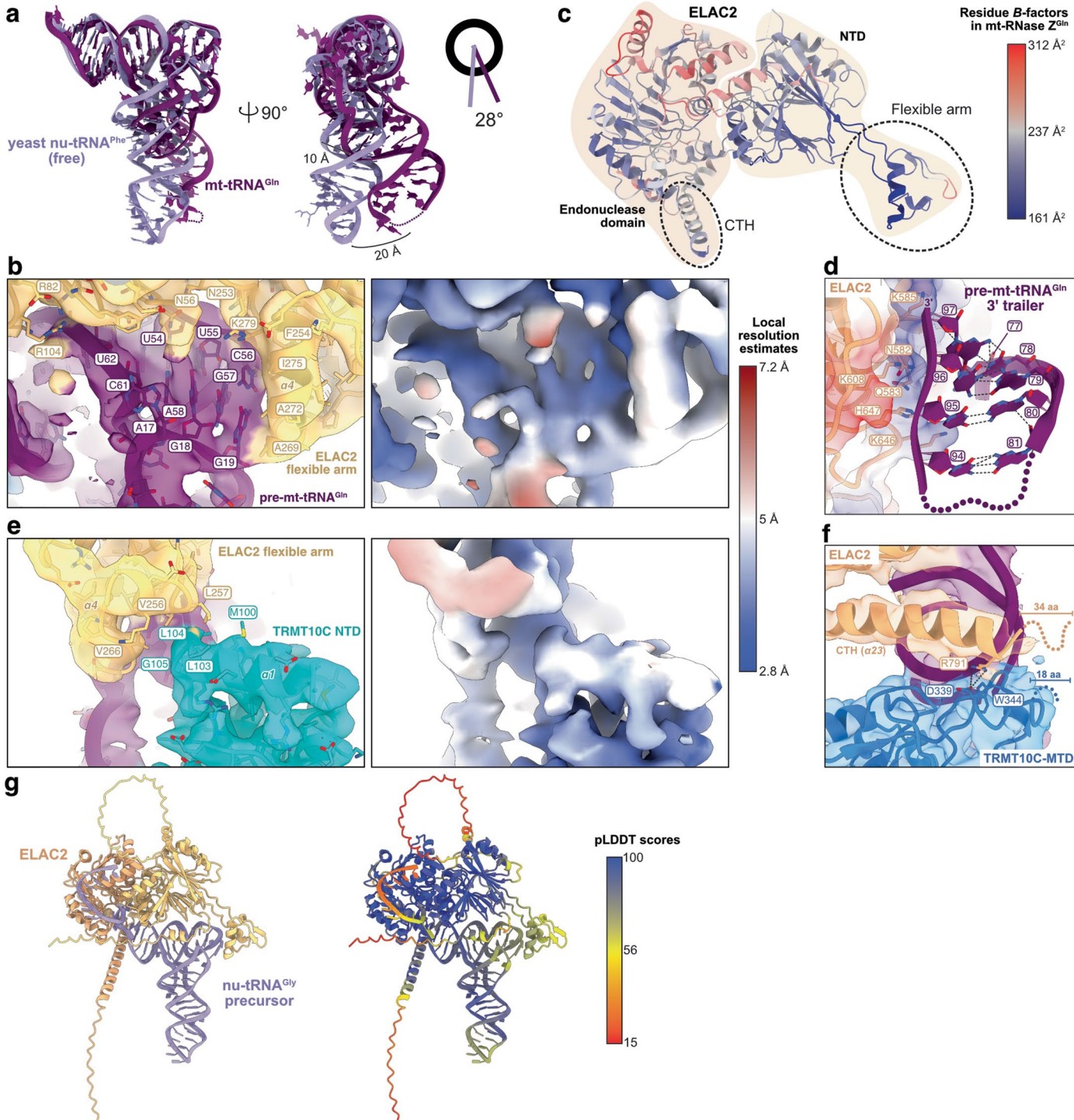

**Extended Data Fig. 6 | Recognition of canonical pre-mt-tRNA by ELAC2. a.** Structural comparison of mt-RNase Z-bound human mt-tRNA$^{Gln}$ and free yeast nu-tRNA$^{Phe}$ (PDB: 4TNA)[62]. **b.** Electron potential map (left) and corresponding local resolution estimates (right) for mt-RNase Z$^{Gln}$ complex at the tRNA elbow–ELAC2 flexible arm interface. Residue sidechains are shown as sticks. Residues within 4 Å of the mt-tRNA$^{Gln}$ elbow–ELAC2 flexible arm interface are labelled. **c.** Cartoon representation of ELAC2 in the mt-RNase Z$^{Gln}$ complex colored by per-residue $B$-factors. The flexible arm residues have the lowest $B$-factor values among ELAC2 residues. **d.** Interactions of the helical segment of the 3′-trailer of pre-mt-tRNA$^{Gln}$ with a positively charged patch in the ELAC2 endonuclease domain. **e.** Electron potential map (left) and corresponding local resolution estimates (right) for mt-RNase Z$^{Gln}$ complex at the TRMT10C NTD–ELAC2 flexible arm interface. Residue sidechains are shown as sticks. Residues within 4 Å of the TRMT10C NTD–ELAC2 flexible arm interface are labelled. **f.** Interactions between ELAC2-CTH and TRMT10C-MTD in the mt-RNase Z$^{Gln}$ complex. 'aa' is short for amino acids. **g.** Alphafold3 structural prediction of the ELAC2-bound nu-tRNA$^{Gly}$ precursor. The model on the right is colored by the per-residue predicted local distance difference test (pLDDT) scores.

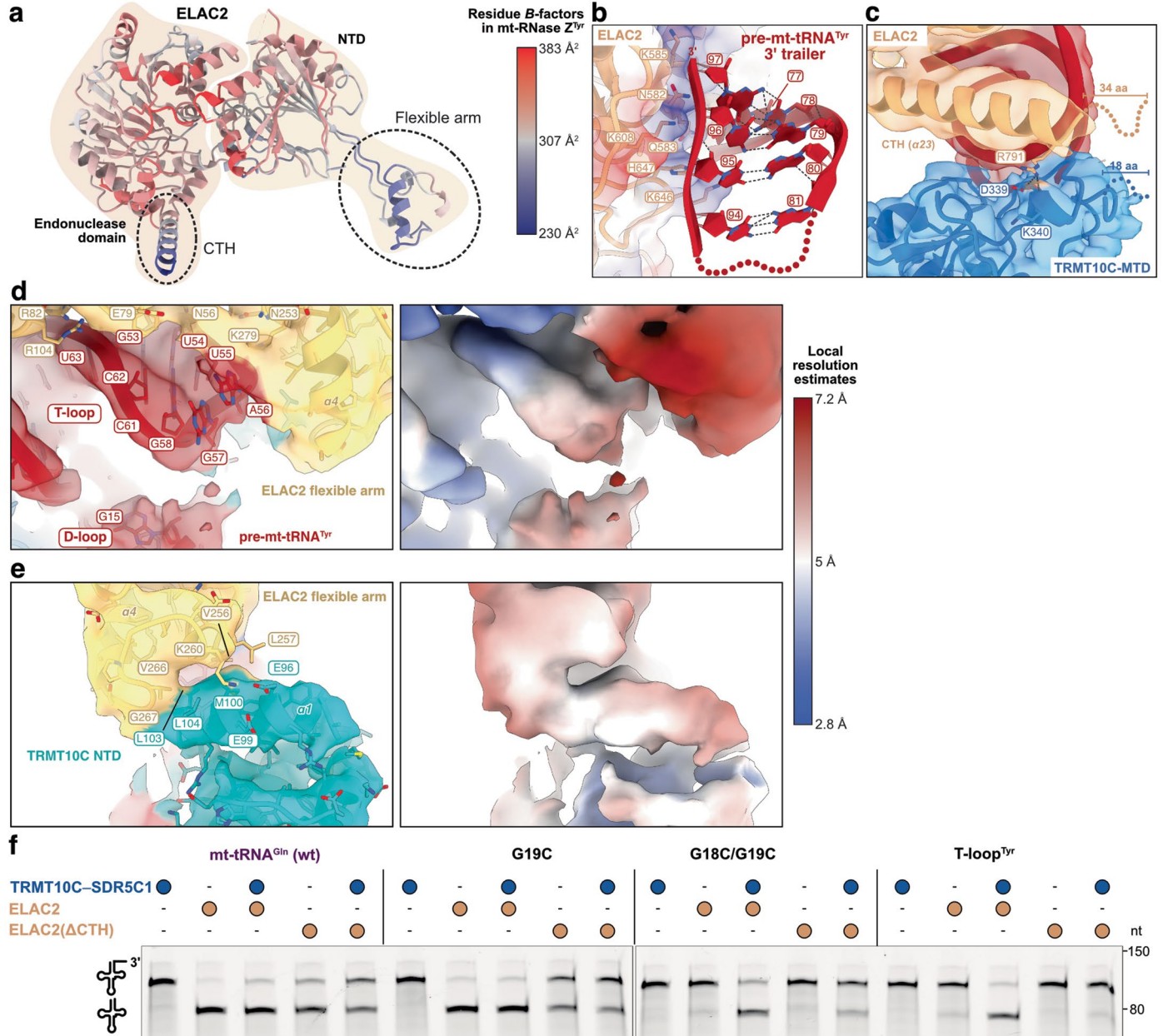

**Extended Data Fig. 7 | Recognition of non-canonical pre-mt-tRNA by ELAC2.**
**a**. Cartoon representation of ELAC2 in mt-RNase Z$^{Tyr}$ complex colored by per-residue *B*-factors. The flexible arm residues have the lowest *B*-factor values among ELAC2 residues. **b**. Interactions of the helical segment of the 3′-trailer of pre-mt-tRNA$^{Tyr}$ with a positively charged patch in the ELAC2 endonuclease domain. **c**. Interactions between ELAC2-CTH and TRMT10C-MTD in mt-RNase Z$^{Tyr}$ complex. 'aa' is short for amino acids. **d**. Electron potential map (left) and corresponding local resolution estimates (right) for mt-RNase Z$^{Tyr}$ complex at the tRNA elbow–ELAC2 flexible arm interface. Residue sidechains are shown as sticks. Residues within 4 Å of the mt-tRNA$^{Tyr}$ elbow–ELAC2 flexible arm interface are labelled. **e**. Electron potential map (left) and corresponding local resolution estimates (right) for mt-RNase Z$^{Tyr}$ complex at the TRMT10C NTD–ELAC2 flexible arm interface. Residue sidechains are shown as sticks. Residues within 4 Å of the TRMT10C NTD–ELAC2 flexible arm interface are labelled. **f**. *In vitro* cleavage assay demonstrating the role of ELAC2-CTH in 3′-processing of tRNAs. 3′-processing of mt-tRNA$^{Gln}$ variants with destabilized elbows is affected notably more than the wild-type mt-tRNA$^{Gln}$ with a stable canonical elbow. Gels are representatives of three independent replicates.

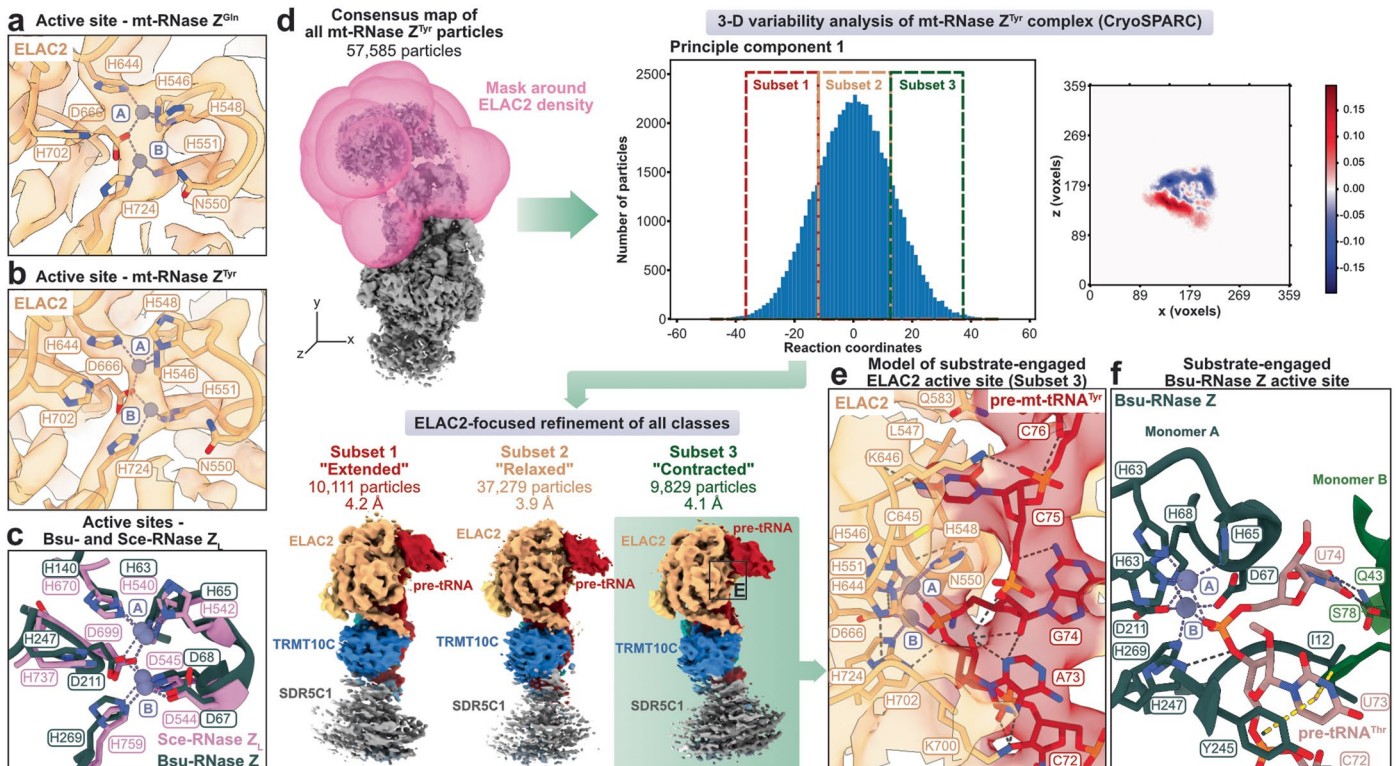

**Extended Data Fig. 8 | Catalytic mechanism of ELAC2. a.** Active site of ELAC2 in the mt-RNase Z$^{Gln}$ complex. The cryo-EM density map is overlaid. **b.** Active site of ELAC2 in mt-RNase Z$^{Tyr}$ complex. The cryo-EM density map is overlaid as in (a). **c.** Active sites of *Bacillus subtilis (Bsu)*-RNase Z (PDB: 4GCW)[52] and *Saccharomyces cerevisiae (Sce)*-RNase Z$_L$ (PDB: 5MTZ)[64]. **d.** 3-D variability analysis of the mt-RNase Z$^{Tyr}$ complex. The "extended", "relaxed" and "contracted" conformational states

refer to the conformation of the pre-tRNA near the 3'-cleavage site. The structures shown here represent snapshots in a continuum of variability. **e.** Model of the substrate-engaged ELAC2 active site based on the cryo-EM density map of the "contracted" state. **f.** Structure of substrate-engaged *Bsu*-RNase Z active site (PDB: 4GCW)[65].

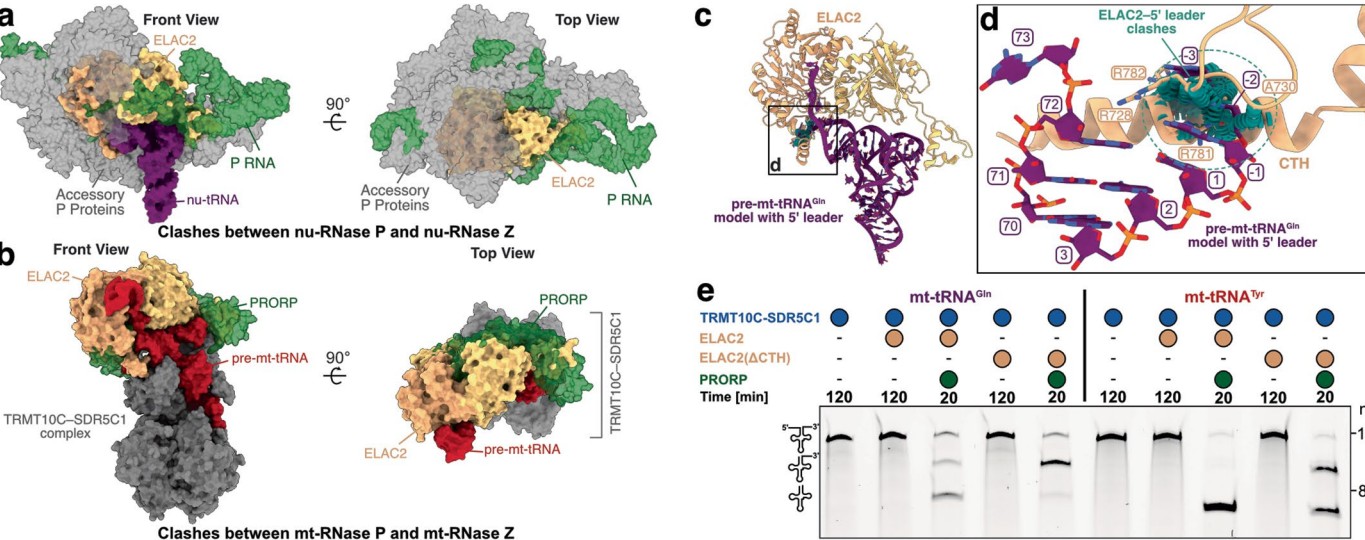

**Extended Data Fig. 9 | Sequential order of tRNA maturation, related to Fig. 2. a**. Overlay of nu-RNase P (PDB: 6AHU)[42] and ELAC2 in complex with tRNA. Nu-RNase P is shown as transparent surface, with the P RNA shown in green and accessory proteins shown in grey. Binding of the two enzymes is mutually exclusive. **b**. Overlay of the mt-RNase P[45] and mt-RNase Z complexes with tRNA. The PRORP subunit of mt-RNase P is shown as a transparent green surface. Binding of PRORP and ELAC2 to the TRMT10C–SDR5C1–pre-tRNA complex is

mutually exclusive. **c**. Structural model of ELAC2 with pre-mt-tRNA(Gln) showing clashes between ELAC2 and the 5′-leader. Region shown in detail in d is indicated. **d**. Clashes between ELAC2 and modelled 5′-leader of pre-mt-tRNA(Gln). The view shown is rotated from c for clarity. Atom-to-atom clashes are shown in teal. **e**. *In vitro* cleavage assay showing that the ELAC2-CTH is not the sole determinant of 5′–3′ tRNA processing order.

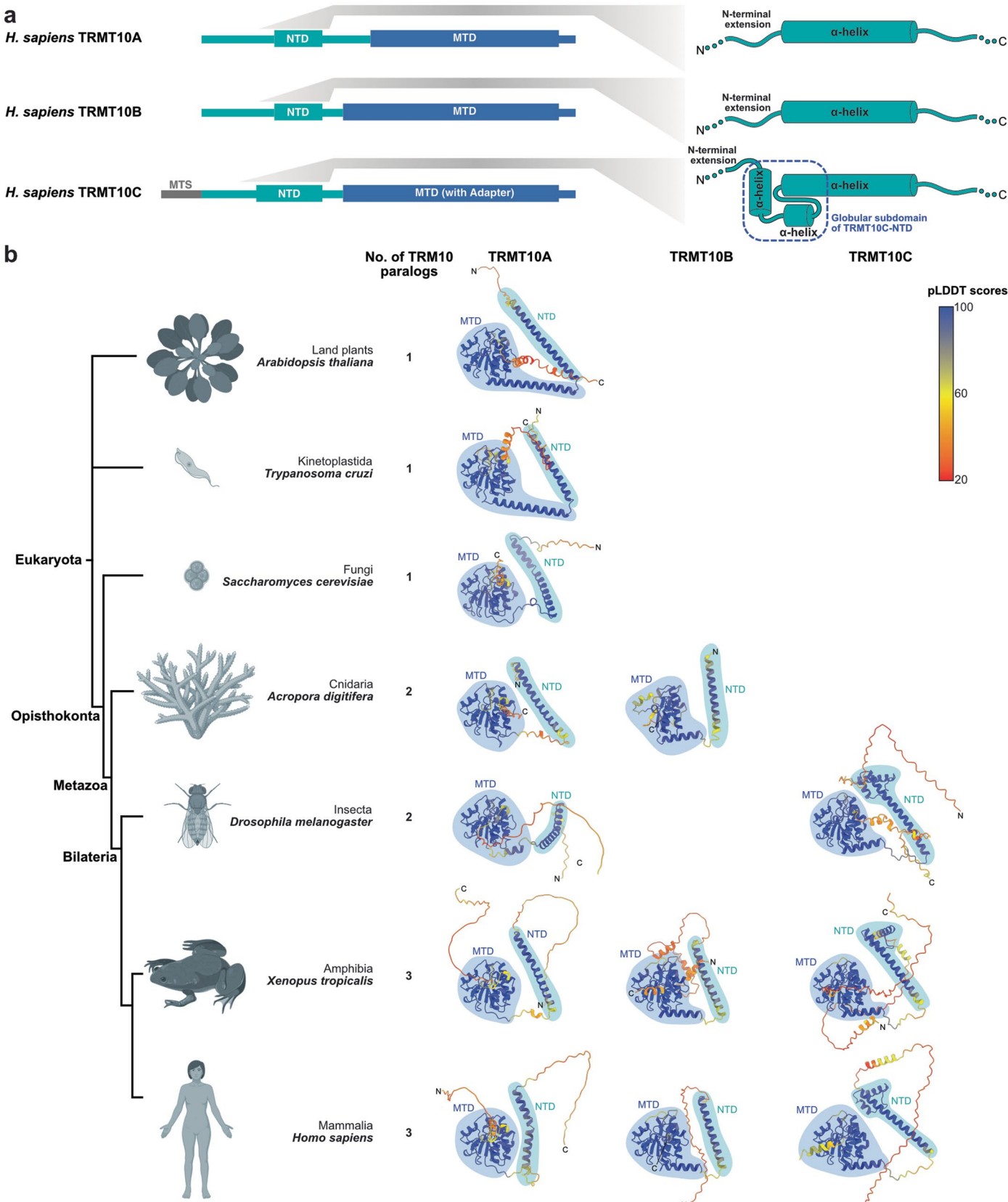

**Extended Data Fig. 10 | TRM10 homologs across the eukaryotic domain of life.**
**a.** Domain architectures of human TRM10 homologs (left) and the secondary structural architecture of their respective NTDs (right; indicated by shaded gray areas). The NTD of TRMT10A and TRMT10B comprises a single long α-helix, while the TRMT10C-NTD is comprised of three α-helices that fold into a globular subdomain. **b.** Comparison of Alphafold-predicted structures of TRM10 homologs across different eukaryotic taxa. The structures are colored by per-residue predicted local distance difference test (pLDDT) scores. The N-terminal domain (NTD) and methyltransferase domain (MTD) are marked and colored in cyan and blue, respectively. TRMT10C, with the globular subdomain in its NTD, is present only in bilaterian animals, coinciding with the structural erosion of their mitochondrial tRNAs[30]. Created with BioRender.com.

# Reporting Summary

## Statistics

For all statistical analyses, confirm that the following items are present in the figure legend, table legend, main text, or Methods section.

| n/a | Confirmed | |
|---|---|---|
| ☐ | ☒ | The exact sample size (*n*) for each experimental group/condition, given as a discrete number and unit of measurement |
| ☐ | ☒ | A statement on whether measurements were taken from distinct samples or whether the same sample was measured repeatedly |
| ☒ | ☐ | The statistical test(s) used AND whether they are one- or two-sided<br>*Only common tests should be described solely by name; describe more complex techniques in the Methods section.* |
| ☐ | ☐ | A description of all covariates tested |
| ☒ | ☐ | A description of any assumptions or corrections, such as tests of normality and adjustment for multiple comparisons |
| ☐ | ☒ | A full description of the statistical parameters including central tendency (e.g. means) or other basic estimates (e.g. regression coefficient) AND variation (e.g. standard deviation) or associated estimates of uncertainty (e.g. confidence intervals) |
| ☒ | ☐ | For null hypothesis testing, the test statistic (e.g. *F*, *t*, *r*) with confidence intervals, effect sizes, degrees of freedom and *P* value noted<br>*Give P values as exact values whenever suitable.* |
| ☒ | ☐ | For Bayesian analysis, information on the choice of priors and Markov chain Monte Carlo settings |
| ☒ | ☐ | For hierarchical and complex designs, identification of the appropriate level for tests and full reporting of outcomes |
| ☒ | ☐ | Estimates of effect sizes (e.g. Cohen's *d*, Pearson's *r*), indicating how they were calculated |

*Our web collection on statistics for biologists contains articles on many of the points above.*

## Software and code

Policy information about availability of computer code

| Data collection | Serial EM v4.0, SPARKCONTROL v3.1 |
|---|---|
| Data analysis | Warp v1.0.9, CryoSPARC v4.2.1, RELION v3.1, WinCOOT v0.9.8.7, PHENIX v1.20, ChimeraX 1.6.1 with Isolde, Prism v10.2.3, PyMOL v2.0 |

For manuscripts utilizing custom algorithms or software that are central to the research but not yet described in published literature, software must be made available to editors and reviewers. We strongly encourage code deposition in a community repository (e.g. GitHub). See the Nature Portfolio guidelines for submitting code & software for further information.

## Data

Policy information about availability of data

All manuscripts must include a data availability statement. This statement should provide the following information, where applicable:
- Accession codes, unique identifiers, or web links for publicly available datasets
- A description of any restrictions on data availability
- For clinical datasets or third party data, please ensure that the statement adheres to our policy

The uniprot accession IDs for ELAC2, TRMT10C and SDR5C1 are Q9BQ52, Q7L0Y3 and Q99714. The structure coordinates for mt-RNase PTyr, Gly-RS-bound nu-tRNAGly, yeast tRNAPhe and human nu-RNase P were obtained from Protein Data Bank (PDB) under accession codes 7ONU, 5E6M, 4TNA and 6AHU. The cryo-EM density reconstructions for mt-RNase ZGln and mt-RNase ZTyr were deposited with the Electron Microscopy Database (EMDB) under accession codes EMD-19455

and EMD-19457. The respective structure coordinates were deposited with the Protein Data Bank (PDB) under accession codes 8RR3 and 8RR4. Source data are provided with this manuscript.

## Research involving human participants, their data, or biological material

Policy information about studies with human participants or human data. See also policy information about sex, gender (identity/presentation), and sexual orientation and race, ethnicity and racism.

| | |
|---|---|
| Reporting on sex and gender | N/A |
| Reporting on race, ethnicity, or other socially relevant groupings | N/A |
| Population characteristics | N/A |
| Recruitment | N/A |
| Ethics oversight | N/A |

Note that full information on the approval of the study protocol must also be provided in the manuscript.

## Field-specific reporting

Please select the one below that is the best fit for your research. If you are not sure, read the appropriate sections before making your selection.

☒ Life sciences          ☐ Behavioural & social sciences          ☐ Ecological, evolutionary & environmental sciences

For a reference copy of the document with all sections, see nature.com/documents/nr-reporting-summary-flat.pdf

## Life sciences study design

All studies must disclose on these points even when the disclosure is negative.

| | |
|---|---|
| Sample size | No statistical methods were used to predetermine sample size for cryo-EM analysis. The required number of micrographs was determined based on particle density in the micrographs to obtain 10E6 to 10E7 total particle images, which is generally sufficient to obtain particle subsets resulting in cryo-EM maps of 2.5-4.0 Angstrom resolution. |
| Data exclusions | No data were excluded from the analysis. |
| Replication | For in vitro cleavage and FA experiments, at least two independent replicates were performed and all performed replicated were successful. |
| Randomization | Cryo-EM particle sets were randomly divided into half-sets and processed independently, as is the standard approach in cryo-EM. |
| Blinding | The investigators were not blinded to allocation during experiments and outcome assessment as this is standard practice in cryo-EM. |

## Reporting for specific materials, systems and methods

We require information from authors about some types of materials, experimental systems and methods used in many studies. Here, indicate whether each material, system or method listed is relevant to your study. If you are not sure if a list item applies to your research, read the appropriate section before selecting a response.

### Materials & experimental systems

| n/a | Involved in the study |
|---|---|
| ☒ | ☐ Antibodies |
| ☐ | ☒ Eukaryotic cell lines |
| ☒ | ☐ Palaeontology and archaeology |
| ☒ | ☐ Animals and other organisms |
| ☒ | ☐ Clinical data |
| ☒ | ☐ Dual use research of concern |
| ☒ | ☐ Plants |

### Methods

| n/a | Involved in the study |
|---|---|
| ☒ | ☐ ChIP-seq |
| ☒ | ☐ Flow cytometry |
| ☒ | ☐ MRI-based neuroimaging |

## Eukaryotic cell lines

Policy information about cell lines and Sex and Gender in Research

| | |
|---|---|
| Cell line source(s) | Hi5 cells: Expression Systems, Tni Insect cells in ESF921 media, item 94-002F<br>Sf9 cells: ThermoFisher, Catalogue Number 12659017, Sf9 cells in Sf-9000TM III SFM Sf21 cells: Expression Systems, SF21 insect cells in ESF921 medium, Item 94-003F |
| Authentication | None of the cell lines were authenticated. |
| Mycoplasma contamination | Cell lines were not tested for mycoplasma contamination. |
| Commonly misidentified lines<br>(See ICLAC register) | No commonly misidentified cell lines were used. |

## Plants

| | |
|---|---|
| Seed stocks | N/A |
| Novel plant genotypes | N/A |
| Authentication | N/A |

