## [Peer Review File · Nature Structural & Molecular Biology]

Molecular basis of human nuclear and mitochondrial tRNA 3'-processing

Corresponding Author: Professor Hauke Hillen

This manuscript has been previously reviewed at another journal. This document only contains information relating to versions considered at Nature Structural & Molecular Biology.

Version 0:

Decision Letter:

9th May 2024

Dear Professor Hillen,

Thank you again for submitting your manuscript "Molecular basis of human nuclear and mitochondrial tRNA 3'-processing". I apologise for the slight delay in responding, which resulted from the difficulty in timely obtaining suitable referee reports. Nevertheless, we now have comments (below) from the 3 reviewers who evaluated your paper. In light of these reports, we remain interested in your study and would like to see your response to the comments of the referees, in the form of a revised manuscript.

You will see that all referees are appreciative of the insight imparted by the work, its conceptual clarity and potential implications. However, the experts additionally raise several notable concerns that need to be addressed in a revised manuscript. More specifically, Reviewer #1 requests additional biochemical experiments, whereas Reviewer #3 notes the need for supporting SPR or TSA assays. Furthermore, all experts point towards needed clarifications, textually addressing potential conceptual concerns, adding important missing information, expanding the introduction and discussion to help the reader, notable clarifications in places where the experimental rationale is unclear, and better presentation/visualization of the data.

Please be sure to address/respond to all concerns of the referees in full in a point-by-point response and highlight all changes in the revised manuscript text file. If you have comments that are intended for editors only, please include those in a separate cover letter.

We expect to see your revised manuscript within 3 months. If you cannot send it within this time, please contact us to discuss an extension; we would still consider your revision, provided that no similar work has been accepted for publication at NSMB or published elsewhere.

Reporting Summary:
<https://www.nature.com/documents/nr-reporting-summary.pdf>

When submitting the revised version of your manuscript, please pay close attention to our

[href="https://www.nature.com/nature-portfolio/editorial-policies/image-integrity">Digital Image Integrity Guidelines.](https://www.nature.com/nature-portfolio/editorial-policies/image-integrity) and to the following points below:

If there are additional or modified structures presented in the final revision, please submit the corresponding PDB validation reports, maps and models.

Data availability: this journal strongly supports public availability of data. All data used in accepted papers should be available via a public data repository, or alternatively, as Supplementary Information. If data can only be shared on request, please explain why in your Data Availability Statement, and also in the correspondence with your editor. Please note that for some data types, deposition in a public repository is mandatory - more information on our data deposition policies and available repositories can be found below:

<https://www.nature.com/nature-research/editorial-policies/reporting-standards#availability-of-data>

Nature Structural & Molecular Biology is committed to improving transparency in authorship. As part of our efforts in this direction, we are now requesting that all authors identified as 'corresponding author' on published papers create and link their Open Researcher and Contributor Identifier (ORCID) with their account on the Manuscript Tracking System (MTS), prior to acceptance. This applies to primary research papers only. ORCID helps the scientific community achieve unambiguous attribution of all scholarly contributions. You can create and link your ORCID from the home page of the MTS by clicking on 'Modify my Springer Nature account'. For more information please visit [please visit www.springernature.com/orcid](http://www.springernature.com/orcid).

Link Redacted

Sincerely,

Dimitris Typas
Senior Editor
Nature Structural & Molecular Biology
ORCID: 0000-0002-8737-1319

Reviewers' Comments:

Reviewer #1:
Remarks to the Author:

The manuscript by Bhatta et al., "Molecular basis of human nuclear and mitochondrial tRNA 3'-processing" presents two cryo-EM structures of tRNA 3' end processing complexes that include pre-tRNA, ELAC2, TRMT10C, and SDR5C1. The manuscript combines the structural information with biochemical findings to show that mt-tRNAs with canonical folds (with elbow) and noncanonical folds (more degenerate without intact elbow) interact with the processing complex differently. Together, their work supports the notion that TRMT10C-SDR5C1 generally aids with reconstituting the tRNA structure that eroded through evolution, especially for mitochondrial tRNA genes.

The biochemical differences among the canonical and noncanonical tRNAs are clear, and the differences in tRNA structures in the elbow region provide strong evidence for the distinction between the tRNAs. Overall, the manuscript is well-written and communicates the concepts clearly. However, the mechanistic explanations in the manuscript need additional clarification and would benefit from some additional data.

1. If canonical tRNA can be processed without TRMT10C-SDR5C1, why did the authors determine the structure with those factors? Can the authors provide a better explanation of why those two complexes were chosen for structure determination?
2. The biochemical assays are single end-point assays. Do the authors have evidence that TRMT10C-SDR5C1 enhances ELAC2-tRNA (canonical) interactions? Providing a more complete picture of their role in processing would be helpful instead of dismissing them as dispensable.
3. The rationale for sequential processing of 5' and 3' ends of tRNA is confusing in the current manuscript. Can the authors also explain why they used the substrate with variable 5' leader in the two structures? If sequential, why was 5' leader included in the first place?
4. The authors state: "While canonical nu-tRNAs and mt-tRNAs are recognized by direct ELAC2-RNA interactions, processing of non-canonical mt-tRNAs depends on protein-protein interactions between ELAC2 and TRMT10C." This model sounds too simplistic because there are still protein-protein interactions in both complexes. The biochemical tests were performed with large deletions, only for enzymatic assays at single concentrations. Can the authors use point mutations to decouple the two effects of TRMT10C-SDR5C1: 1) supporting tRNA structure and 2) increasing affinity for ELAC2? For example, could they help the tRNAs to remain folded well enough so that they can aid with ELAC2 processing even when some of the protein-protein contacts are weakened?
5. Just curious, are ELAC2 affinities for the canonical and noncanonical tRNAs different?

Reviewer #2:

Remarks to the Author:

The precursor of human mitochondrial (mt) tRNAs undergoes multiple processes to mature into functional forms. The 5'-leader and 3'-tail sequences of precursor mt tRNA are cleaved by mt endonucleases RNaseP and RNaseZ, respectively. Subsequently, the 3'-CCA sequence is added by the mt CCA-adding enzyme. Throughout these processes, nucleotide modifications are introduced to mt tRNAs, rendering them functional. Some enzymes are shared between the maturation processes of nuclear (nc) and mt tRNAs. RNaseZ is involved in 3'-tail removal from precursor mt tRNAs and nc tRNAs. Human mt tRNA structures deviate from nc tRNA structures in the elbow region (interaction between D- and TpsiC-arms). In mt tRNAs, the sequences in the D- and TpsiC-arms are not well conserved and are distinct from those in nctRNAs. In a particular case, tRNA lacks the D-arm (tRNASerAGY). Therefore, efficient processing of mt tRNAs requires additional factors. For the 3'-end processing of mt tRNAs by RNaseZ, TRMT10C and SDR5C1 are necessary, while 3'-end processing of nc tRNAs is catalyzed by RNaseZ alone. Hence, the requirement of TRMT10C and SDR5C1 for mt tRNA 3'-processing by RNaseZ remains elusive.

In the manuscript entitled "Molecular Basis of Human Nuclear and Mitochondrial tRNA 3'-Processing" by Hauke Hillen et al., the molecular basis of tRNA 3'-processing by RNaseZ, particularly the necessity of TRMT10C and SDR5C1 for mt tRNA 3'-processing, is clearly presented. For mt tRNA 3'-processing, direct interaction between ELAC2 and TRMT10C is required to compensate for the structural erosion of mt tRNAs.

Last year, the group led by Dr. Carine Tisne (IBPC, CNRS) published a pre-print (Structural basis of human mitochondrial maturation, doi: <https://doi.org/10.1101/2023.12.19.572246>). In the pre-print, the authors present the structures representing pre-mt tRNA 5'-processing, 3'-processing, and 3'-CCA addition using TRMT10C and SDR5C1 as the platform for the reaction.

However, the present study by Hauke Hillen et al., specifically focuses on the differences in 3'-processing of pre-nuclear tRNA and pre-mt-tRNA, thus the report from the other group does not diminish the novelty of the present study at all.

This reviewer enjoyed reviewing this manuscript and appreciates the structural insights into mitochondrial tRNA 3'-processing machinery. The biochemical and structural analyses in this manuscript are solid. The manuscript is well organized and presented. The present study provides not only the detailed mechanism of mt tRNA 3'-processing by RNaseZ-TRMT10C-SDR5C1 complex, but also expands our understanding of how the function or structure of RNAs are compensated by proteins in mitochondrial system, where rRNA and tRNA are degenerated.

This reviewer strongly recommends the publication of the study in Nature Structural Molecular Biology and has few minor comments.

Minor points

- i) In some metazoan mitochondria, mt tRNAs lack D-arm, T-arm, or both. In mammals, the tRNASer for the AGY codon lacks the D-arm. The authors should kindly discuss the 3'-processing mechanism of tRNASerAGY in the revised manuscript.
- ii) Related to i), furthermore, in nematodes, most mt RNAs lack both the D- and T-regions. How does mt tRNA 3'-processing

proceed? In nematodes, RNaseZ and TRMT10C differ from those in mammals? Are additional enzyme(s) used for the process?

iii) Authors could kindly cite the paper by Ohtsuki et al. (DOI: 10.1038/nsb826) "A unique serine-specific elongation factor Tu found in nematode mitochondria" and further discuss the structural erosion of mt RNAs. This may deepen the biological relevance of the present study.

Reviewer #3:

Remarks to the Author:

In this manuscript, Arjun Bhatta and colleagues have developed a combination of in vitro maturation assays of pre-tRNA, with cryo-EM analyses of mitochondrial tRNA 3'-end processing complexes extended to nuclear ELAC2-tRNA complexes (for the in vitro processing analysis). This clever and elegant work allow the authors to propose original explanations about the precise role of TRMT10C and SDR5C1 in the 3'-end processing of mt-tRNA, and why they are crucial for the recognition and maturation of 'degenerated' mt-tRNA. Overall, the manuscript is well written and provides key elements to be understood by a broad readership. The scientific investigation herein described is consequent, solid and very convincing. Beyond the tRNA maturation process which is the primary scope of this manuscript, the interpretation of the results allows this study to propose a new and interesting response to a general question, which is what is exactly the role of helper proteins that have no known enzymatic functions, and are often designated as "recruitment platforms" or "structural stunts" within a complex processing reaction. I therefore think that this manuscript is worth publishing in NSMB, provided that the comments below are addressed.

Nuclear tRNA 3'-end recognition and processing by ELAC2 alone (Figure 4): based on their cryo-EM analysis of mt-tRNA processing, the authors propose a model for nuclear tRNA 3' end maturation by ELAC2 alone. However, although supported by processing assays performed in vitro, this remains a bit speculative. What is known about in vivo mechanisms of ELAC2 action for nu-tRNA maturation? Does it need accessory helper proteins? To reply these questions, one solution could be to purify nu-RNaseZGly from human cells and characterize their structure and composition by cryo-EM and MS, but this is out of the scope of the current manuscript. If no further functional proofs of this model can be brought, the conclusions drawn in this paragraph should be toned down, and maybe merged with the final results paragraph "Structural basis for sequential tRNA processing in mitochondria and nucleus". Along the same line, although the introduction gives the naïve reader key elements to understand the importance of this work, it might be worth giving a few more details about the full processing pathway of mt-tRNA (maybe as a figure?) to better grasp how complex it is, and what are the common points and differences between nuclear and mitochondrial tRNA maturation schemes (cellular localization, required factors, in vitro or in vivo results...)

Figure 1a : to be consistent, and for a broad audience, keep the 3-letter code describing amino acids instead of single-letter one.

Figure 1b: The difference between conserved and degenerated mt-tRNA is clearly visible in the 2D representation provided, could the authors display the 3D structures of such different mt-tRNA, to visualize how these differences translate in space?

Figure 1b: Inhibition of 3'-end cleavage by TRMT10C-SRDC1 : what is the interpretation of the authors of this observation?

In figure 1d, what is the band above pre-mt-tRNAGln (and to some extent pre-mt-tRNATyr)? Is there other cleavage sites/partial pre-cleavages occurring in these pre-mt-tRNA in vitro?

Cryo-EM structures of mt-RNase ZTyr and mt-RNase ZGln complexes (Figure 2): is one of the complexes more stable than the other? From cryo-EM processing & resolution reached, one structure appears more flexible than the other, but could the authors implement TSA or SPR assays to further explore whether the binding affinity of ELAC2 and/or TRMT10C for mt-TRNATyr or mt-tRNAGln are different?

Figure 3d, f and 5, d,f: local resolution of these parts of the maps are not entirely clear from the provided supplementary data. Please show the cryo-EM density around the atomic models to be more convincing (in supplementary data panels for instance)

Version 1:

Decision Letter:

Our ref: NSMB-A49059A

26th Jul 2024

Dear Professor Hillen,

Thank you for submitting your revised manuscript "Molecular basis of human nuclear and mitochondrial tRNA 3'-processing" (NSMB-A49059A). It has now been seen by the original referees and their comments are below. The reviewers find that the paper has improved in revision, and therefore we'll be happy to accept it in principle in Nature Structural & Molecular

Biology, pending minor revisions to satisfy the referees' final requests and to comply with our editorial and formatting guidelines.

We are now performing detailed checks on your paper and will send you a checklist detailing our editorial and formatting requirements in about 2-4 weeks. Please do not upload the final materials and make any revisions until you receive this additional information from us.

Thank you again for your interest in Nature Structural & Molecular Biology. Please do not hesitate to contact me if you have any questions.

Sincerely,

Dimitris Typas
Senior Editor
Nature Structural & Molecular Biology
ORCID: 0000-0002-8737-1319

Reviewer #1 (Remarks to the Author):

The revised version is acceptable for publication. I only have a minor comment. The methods should contain pH for each buffer. Some are missing, such as for the final cryo-EM sample before freezing the grids and for purifying the RNA.

Reviewer #2 (Remarks to the Author):

The authors revised the manuscript, and the manuscript is now acceptable for publication in NSMB.

Reviewer #3 (Remarks to the Author):

Dr. Hillen and colleagues have integrated all feedback from the three reviewers into their revised manuscript. All of my concerns have been adequately addressed. I have no further comments to make, other than to note that the work presented here is of a high standard and can be published in Nature Structural and Molecular Biology in its current form.

Version 2:

Decision Letter:

6th Nov 2024

Dear Professor Hillen,

We are now happy to accept your revised paper "Molecular basis of human nuclear and mitochondrial tRNA 3'-processing" for publication as an Article in Nature Structural & Molecular Biology.

To assist our authors in disseminating their research to the broader community, our SharedIt initiative provides all co-authors with the ability to generate a unique shareable link that will allow anyone (with or without a subscription) to read the

published article. Recipients of the link with a subscription will also be able to download and print the PDF.

Your paper will be published online soon after we receive proof corrections and will appear in print in the next available issue. You can find out your date of online publication by contacting the production team shortly after sending your proof corrections.

Please note that *Nature Structural & Molecular Biology* is a Transformative Journal (TJ). Authors may publish their research with us through the traditional subscription access route or make their paper immediately open access through payment of an article-processing charge (APC). Authors will not be required to make a final decision about access to their article until it has been accepted. [Find out more about Transformative Journals](https://www.springernature.com/gp/open-research/transformative-journals)

Authors may need to take specific actions to achieve [compliance with funder and institutional open access mandates](https://www.springernature.com/gp/open-research/funding/policy-compliance-faqs). If your research is supported by a funder that requires immediate open access (e.g. according to [Plan S principles](https://www.springernature.com/gp/open-research/plan-s-compliance)) then you should select the gold OA route, and we will direct you to the compliant route where possible. For authors selecting the subscription publication route, the journal's standard licensing terms will need to be accepted, including [self-archiving policies](https://www.springernature.com/gp/open-research/policies/journal-policies). Those licensing terms will supersede any other terms that the author or any third party may assert apply to any version of the manuscript.

Sincerely,

Dimitris Typas
Senior Editor
Nature Structural & Molecular Biology
ORCID: 0000-0002-8737-1319

Molecular basis of human nuclear and mitochondrial tRNA 3'-processing

Bhatta, Kuhle, Yu et al.

Response to reviewer comments

Responses are in blue italics

Reviewer #1:

Remarks to the Author:

The manuscript by Bhatta et al., "Molecular basis of human nuclear and mitochondrial tRNA 3'-processing" presents two cryo-EM structures of tRNA 3' end processing complexes that include pre-tRNA, ELAC2, TRMT10C, and SDR5C1. The manuscript combines the structural information with biochemical findings to show that mt-tRNAs with canonical folds (with elbow) and noncanonical folds (more degenerate without intact elbow) interact with the processing complex differently. Together, their work supports the notion that TRMT10C-SDR5C1 generally aids with reconstituting the tRNA structure that eroded through evolution, especially for mitochondrial tRNA genes.

The biochemical differences among the canonical and noncanonical tRNAs are clear, and the differences in tRNA structures in the elbow region provide strong evidence for the distinction between the tRNAs. Overall, the manuscript is well-written and communicates the concepts clearly. However, the mechanistic explanations in the manuscript need additional clarification and would benefit from some additional data.

We thank the reviewer for their positive feedback and constructive comments, which we have addressed as outlined below.

1. If canonical tRNA can be processed without TRMT10C-SDR5C1, why did the authors determine the structure with those factors? Can the authors provide a better explanation of why those two complexes were chosen for structure determination?

The goal of our approach was to determine whether mt-RNase Z is a multi-subunit complex containing TRMT10C and SDR5C1 even for canonical tRNAs. While we clearly show that canonical tRNAs do not require TRMT10C-SDR5C1 for 3' processing, previous in vivo as well as in vitro experiments suggested that both canonical as well as non-canonical mitochondrial tRNAs require TRMT10C and SDR5C1 for the preceding 5' cleavage step by mt-RNase P (PMIDs 27498866, 29040705). These observations raised the question whether canonical mt-tRNAs are released from the TRMT10C-SDR5C1 complex after 5' cleavage or whether they remain bound to TRMT10C-SDR5C1 during 3' cleavage, even if it is not strictly required. To address this question, we recapitulated the physiological order of events by first performing 5' processing of the canonical mt-tRNA^{Gln} in presence of TRMT10C-SDR5C1 and subsequently adding a catalytic mutant of ELAC2 to stabilize the mt-RNase Z. We then analyzed the resulting stalled 3' processing complex via sucrose density gradient. Here, we observed co-migration of ELAC2, TRMT10C and SDR5C1, suggesting that TRMT10C and SDR5C1 remain bound to mt-tRNA^{Gln} during 3' processing. Thus, our data suggest that TRMT10C and SDR5C1 are part of mt-RNase Z complexes even with canonical mt-tRNAs.

To further substantiate this, we now include additional data in the revised manuscript: 1.) We performed in vitro 5' cleavage assays to confirm the requirement of TRMT10C-SDR5C1 for 5' processing of mt-tRNA^{Gln} (included in the new Extended Data Figure 1d). 2.) We determined the affinity of the TRMT10C-SDR5C1 complex for 5'-processed tRNAs using fluorescence anisotropy (included in the new Extended Data Figure 1c). These experiments show that both canonical and non-canonical tRNAs bind TRMT10C-SDR5C1 with similar affinities, thus providing a mechanistic explanation why both remain bound to TRMT10C-SDR5C1 after 5' processing. We have rephrased the text to clarify our rationale and these data (lines 167-177).

2. The biochemical assays are single end-point assays. Do the authors have evidence that TRMT10C-SDR5C1 enhances ELAC2-tRNA (canonical) interactions? Providing a more complete picture of their role in processing would be helpful instead of dismissing them as dispensable.

We agree with the reviewer on the limitations of the biochemical assay regarding the role of the TRMT10C-SDR5C1 complex for canonical mt-tRNA processing. To obtain more insights into the interactions between the components of the complex, we have therefore additionally performed affinity measurements using fluorescence anisotropy (shown in the new Extended Data Figure 1c). These measurements show that ELAC2 alone has higher affinity for canonical tRNA than to non-canonical tRNA, while the TRMT10C-SDR5C1 complex binds both with similar affinities. These results are consistent with our hypotheses regarding the compensatory role of TRMT10C-SDR5C1 for processing of non-canonical tRNAs, as it facilitates efficient binding of non-canonical substrates. We also attempted to measure whether there is a difference in affinity of ELAC2 to TRMT10C-SDR5C1-bound tRNA substrates. However, as shown in the figure below, we were not able to obtain reliable affinity measurements, likely due to a too small change in mass upon ELAC2-binding to the already very large TRMT10C-SDR5C1-tRNA complex. Thus, we do not have any evidence (or indications) that TRMT10C-SDR5C1 enhances interactions of ELAC2 with canonical tRNAs. We have rephrased the text to reflect this uncertainty by stating that TRMT10C-SDR5C1 is "not strictly required" for processing of canonical tRNAs.

3. The rationale for sequential processing of 5' and 3' ends of tRNA is confusing in the current manuscript. Can the authors also explain why they used the substrate with variable 5' leader in the two structures? If sequential, why was 5' leader included in the first place?

As outlined in the response to the first comment above, our rationale to use a canonical mt-tRNA with a 5' leader to reconstitute the mt-RNase Z complex was to recapitulate the sequential maturation of mt-tRNAs as expected in vivo in order to test whether TRMT10C and SDR5C1, which are required for 5' processing, are retained in the 3' processing complex as well. In the revised manuscript, we have rewritten the text in lines 167-177 to better clarify this and hope this addresses the issue raised by the reviewer adequately.

4. The authors state: "While canonical nu-tRNAs and mt-tRNAs are recognized by direct ELAC2-RNA interactions, processing of non-canonical mt-tRNAs depends on protein-protein interactions between ELAC2 and TRMT10C." This model sounds too simplistic because there are still protein-protein interactions in both complexes. The biochemical tests were performed with large deletions, only for enzymatic assays at single concentrations. Can the authors use point mutations to decouple the two effects of TRMT10C-SDR5C1: 1) supporting tRNA structure and 2) increasing affinity for ELAC2? For example, could they help the tRNAs to remain folded well enough so that they can aid with ELAC2 processing even when some of the protein-protein contacts are weakened?

We agree with the reviewer that the effect of the TRMT10C-SDR5C1 complex on non-canonical mt-tRNA maturation is likely not based just on protein-protein interactions but is likely twofold: 1.) stabilizing the tRNA structure and 2.) supporting ELAC2-binding via direct interactions. Hence, we do not suggest that processing of non-canonical mt-tRNAs depends only on protein-protein interactions between ELAC2 and TRMT10C. However, our biochemical data clearly show that only processing of non-canonical mt-tRNAs critically depends on protein-protein interactions between ELAC2 and TRMT10C NTD (Figure 5g), whereas canonical mt-tRNAs do not (Figure 3g) – even if the mt-RNase Z-mt-tRNA^{Gln} structure suggests that ELAC2 and TRMT10C NTD still make some direct contact. The relative contribution of either effect may also depend on the intrinsic stability of the tRNA fold as well as the availability of the canonical recognition elements, and may thus differ for different tRNAs. In the revised manuscript, we have rephrased the text to emphasize that TRMT10C likely also makes an important contribution to tRNA stabilization in addition to its role in ELAC2 flexible arm binding (lines 109 to 111, 402 to 407).

Regarding the suggestion of point mutants, we agree with the reviewer that it would be interesting to experimentally tease apart the contribution of both effects. Unfortunately, however, we believe that it is not easily possible to decouple the "tRNA structural stabilization" and "ELAC2 engagement" effects of TRMT10C via point mutations as the ELAC2 flexible arm-TRMT10C NTD interface is comprised of multiple, likely redundant, hydrophobic interactions. We initially intended to generate a TRMT10C variant with alanine substitutions in all the NTD side chains involved in interactions with ELAC2. However, this variant (and any other containing point mutations) would not have ensured that the ELAC2-TRMT10C interaction was indeed disrupted specifically, and this cannot be tested easily because the interaction always depends on the presence of a tRNA substrate. We thus instead designed the TRMT10C-ΔNTD mutant (Δ1-124) on the basis that 1.) it completely abolishes any potential interactions with the ELAC2 flexible arm while 2.) still retaining the entire long α-helix (125-155) that interacts with the tRNA's anticodon stem, D-arm and T-arm. Thus, this construct contains the parts of TRMT10C that are responsible for the structural stabilization of the tRNA (which are also conserved in the other TRMT10 homologues TRMT10A and B), but lacks only the TRMT10C-specific globular portion responsible for interactions with ELAC2. We thus consider that our data on the TRMT10C Δ1-124 mutant, in combination with the structural data, are sufficient to show that 1.) stabilization of the non-canonical tRNA per se is not sufficient to allow 3' processing by ELAC2, and 2.) the direct protein-protein interactions between the ELAC2 flexible arm and TRMT10C NTD specifically compensate for loss of ELAC2-tRNA elbow interactions in non-canonical tRNAs. Based on the above reasoning, we believe additional point mutants would not provide more meaningful insights, and hope the reviewer agrees with this.

5. Just curious, are ELAC2 affinities for the canonical and noncanonical tRNAs different?

In response to this reviewer comment, we have tested binding affinities of ELAC2 for canonical (mt-tRNA^{Gln}) and non-canonical (mt-tRNA^{Tyr}) tRNA precursors. We find that TRMT10C-SDR5C1 binds both tRNAs with nearly the same affinity (with K_ds of 167 nM and 215 nM for mt-tRNA^{Gln} and mt-tRNA^{Tyr}, respectively). By contrast, ELAC2

has high affinity only for mt-tRNA^{Gln} (Kd of 301 nM), whereas its affinity for mt-tRNA^{Tyr} is significantly lower (Kd could not be determined accurately; >6 μ M). The new data are included in the revised manuscript in Extended Data Figure 1c and lines 144 to 148.

Reviewer #2:
Remarks to the Author:

The precursor of human mitochondrial (mt) tRNAs undergoes multiple processes to mature into functional forms. The 5'-leader and 3'-tail sequences of precursor mt tRNA are cleaved by mt endonucleases RNaseP and RNaseZ, respectively. Subsequently, the 3'-CCA sequence is added by the mt CCA-adding enzyme. Throughout these processes, nucleotide modifications are introduced to mt tRNAs, rendering them functional. Some enzymes are shared between the maturation processes of nuclear (nc) and mt tRNAs. RNaseZ is involved in 3'-tail removal from precursor mt tRNAs and nc tRNAs.

Human mt tRNA structures deviate from nc tRNA structures in the elbow region (interaction between D- and TpsiC-arms). In mt tRNAs, the sequences in the D- and TpsiC-arms are not well conserved and are distinct from those in nctRNAs. In a particular case, tRNA lacks the D-arm (tRNASerAGY). Therefore, efficient processing of mt tRNAs requires additional factors. For the 3'-end processing of mt tRNAs by RNaseZ, TRMT10C and SDR5C1 are necessary, while 3'-end processing of nc tRNAs is catalyzed by RNaseZ alone. Hence, the requirement of TRMT10C and SDR5C1 for mt tRNA 3'-processing by RNaseZ remains elusive.

In the manuscript entitled "Molecular Basis of Human Nuclear and Mitochondrial tRNA 3'-Processing" by Hauke Hillen et al., the molecular basis of tRNA 3'-processing by RNaseZ, particularly the necessity of TRMT10C and SDR5C1 for mt tRNA 3'-processing, is clearly presented. For mt tRNA 3'-processing, direct interaction between ELAC2 and TRMT10C is required to compensate for the structural erosion of mt tRNAs.

Last year, the group led by Dr. Carine Tisne (IBPC, CNRS) published a pre-print (Structural basis of human mitochondrial maturation, doi:<https://doi.org/10.1101/2023.12.19.572246>). In the pre-print, the authors present the structures representing pre-mt tRNA 5'-processing, 3'-processing, and 3'-CCA addition using TRMT10C and SDR5C1 as the platform for the reaction.

However, the present study by Hauke Hillen et al., specifically focuses on the differences in 3'-processing of pre-nuclear tRNA and pre-mt-tRNA, thus the report from the other group does not diminish the novelty of the present study at all.

This reviewer enjoyed reviewing this manuscript and appreciates the structural insights into mitochondrial tRNA 3'-processing machinery. The biochemical and structural analyses in this manuscript are solid. The manuscript is well organized and presented. The present study provides not only the detailed mechanism of mt tRNA 3'-processing by RNaseZ-TRMT10C-SDR5C1 complex, but also expands our understanding of how the function or structure of RNAs are compensated by proteins in mitochondrial system, where rRNA and tRNA are degenerated.

This reviewer strongly recommends the publication of the study in Nature Structural Molecular Biology and has few minor comments.

We thank the reviewer for their support and valuable suggestions, which we have addressed in the revised manuscript as outlined below.

Minor points

i) In some metazoan mitochondria, mt tRNAs lack D-arm, T-arm, or both. In mammals, the tRNASer for the AGY codon lacks the D-arm. The authors should kindly discuss the 3'-processing mechanism of tRNASerAGY in the revised manuscript.

As suggested, we have now included a paragraph (lines 414 to 421) discussing the mechanism of 3' processing of tRNA^{Ser(AGY)}.

ii) Related to i), furthermore, in nematodes, most mt RNAs lack both the D- and T-regions. How does mt tRNA 3'-processing proceed? In nematodes, RNaseZ and TRMT10C differ from those in mammals? Are additional enzyme(s) used for the process?

We have included lines 462 to 468 discussing the structural erosion in mitochondria-encoded RNAs accompanied by protein-based compensation in metazoans. In nematodes, our sequence analysis and structural prediction suggests that TRMT10C and ELAC2 have undergone domain expansions that have not been characterized but could play a role in recognition/processing of nematode mt-tRNAs. To our knowledge, it is not presently known whether any additional factors are necessary for nematode mt-tRNA processing.

iii) Authors could kindly cite the paper by Ohtsuki et al. (DOI: 10.1038/nsb826) "A unique serine-specific elongation factor Tu found in nematode mitochondria" and further discuss the structural erosion of mt RNAs. This may deepen the biological relevance of the present study.

In the same paragraph addressing comment ii, we now discuss nematode elongation factor Tu 1 and 2 in the context of protein-based compensation of mt-tRNA structural erosion and cite the suggested paper.

Reviewer #3:

Remarks to the Author:

In this manuscript, Arjun Bhatta and colleagues have developed a combination of in vitro maturation assays of pre-tRNA, with cryo-EM analyses of mitochondrial tRNA 3'-end processing complexes extended to nuclear ELAC2-tRNA complexes (for the in vitro processing analysis). This clever and elegant work allow the authors to propose original explanations about the precise role of TRMT10C and SDR5C1 in the 3'-end processing of mt-tRNA, and why they are crucial for the recognition and maturation of 'degenerated' mt-tRNA. Overall, the manuscript is well written and provides key elements to be understood by a broad readership. The scientific investigation herein described is consequent, solid and very convincing. Beyond the tRNA maturation process which is the primary scope of this manuscript, the interpretation of the results allows this study to propose a new and interesting response to a general question, which is what is exactly the role of helper proteins that have no known enzymatic functions, and are often designated as "recruitment platforms" or "structural stunts" within a complex processing reaction. I therefore think that this manuscript is worth publishing in NSMB, provided that the comments below are addressed.

We thank the reviewer for their positive and constructive comments, which we addressed in the revised manuscript as outlined below.

Nuclear tRNA 3'-end recognition and processing by ELAC2 alone (Figure 4): based on their cryo-EM analysis of mt-tRNA processing, the authors propose a model for nuclear tRNA 3' end maturation by ELAC2 alone. However, although supported by processing assays performed in vitro, this remains a bit speculative. What is known about in vivo mechanisms of ELAC2 action for nu-tRNA maturation? Does it need accessory helper proteins? To reply these questions, one solution could be to purify nu-RNaseZGly from human cells and characterize their structure and composition by cryo-EM and MS, but this is out of the scope of the current manuscript. If no further functional proofs of this model can be brought, the conclusions drawn in this paragraph should be toned down, and maybe merged with the final results paragraph "Structural basis for sequential tRNA processing in mitochondria and nucleus".

Indeed, surprisingly little is known about the precise mechanisms of ELAC2-mediated nu-tRNA maturation, in particular regarding its potential interplay with other RNA binding proteins. However, it has previously been shown that ELAC2 is the endonuclease responsible for processing of both nu-tRNAs and mt-tRNAs in vivo, and that it does not require any further factors for nu-tRNA processing (PMID 30126926). While this does not exclude that ELAC2 may interact with other players in vivo that are not directly involved in catalysis, it demonstrates that no other proteins are required for its catalytic activity on nu-tRNAs – in contrast to the situation for mt-tRNAs. In the revised manuscript, we have expanded the introduction to include the reference to the in vivo data for ELAC2 and rephrased to emphasize that the available data suggest that ELAC2 alone appears to be sufficient for nuclear tRNA processing (lines 69 to 71,90 to 93).

As suggested by the reviewer, we have also reorganized the manuscript with regards to the description of nuclear tRNA processing. In the revised manuscript, we have combined our model of nuclear tRNA 3' processing with the preceding section on recognition of canonical mt-tRNAs by ELAC2, as the biochemical results, the structural model and the mechanistic implications for nuclear tRNA processing are conceptually related to this section rather than the final results paragraph on sequential tRNA maturation. Furthermore, we have revised the text in this section (lines 256 to 277) to tone down our conclusions, as the reviewer suggests.

Along the same line, although the introduction gives the naïve reader key elements to understand the importance of this work, it might be worth giving a few more details about the full processing pathway of mt-tRNA (maybe as a figure?) to better grasp how complex it is, and what are the common points and differences between nuclear and mitochondrial tRNA maturation schemes (cellular localization, required factors, in vitro or in vivo results...)

In the revised manuscript, we have expanded the introduction section (lines 64 to 81) to include a broader summary of nuclear and mitochondrial tRNA processing pathways, as suggested by the reviewer. We now provide details on the localization, required factors and general mechanisms of both nuclear and mitochondrial tRNA processing. After careful consideration, we have decided not to include a more detailed introductory figure on this, as it would go beyond the length limitations and scope of an original research article. We do, however, agree that it would be timely to provide a detailed comparison between the two tRNA maturation systems in the light of recent advances, which we think would be best done in a separate review article on the topic that we plan to write in the near future.

Figure 1a: to be consistent, and for a broad audience, keep the 3-letter code describing amino acids instead of single-letter one.

We have revised the figure accordingly.

Figure 1b: The difference between conserved and degenerated mt-tRNA is clearly visible in the 2D representation provided, could the authors display the 3D structures of such different mt-tRNA, to visualize how these differences translate in space?

We have included a schematic depiction which highlights the differences in a three-dimensional model of a tRNA.

Figure 1b: Inhibition of 3'-end cleavage by TRMT10C-SRDC1: what is the interpretation of the authors of this observation?

The slight inhibition of 3'-end cleavage of canonical tRNAs by TRMT10C-SRDC1 is indeed an interesting observation, for which our available data do not provide a clear explanation. We would like to point out, however, that this does not affect our interpretation of canonical mt-tRNA processing by mt-RNase Z. Our biochemical cleavage assays, binding assays (new Extended Data Figure 1c), and in vitro reconstitution experiments suggest that canonical tRNAs are stably bound by TRMT10C-SDR5C1 and remain bound in the presence of ELAC2. As 5' processing of mt-tRNA^{Gln} by mt-RNase P is strictly dependent on TRMT10C-SDR5C1, this suggests that canonical tRNAs remain bound to the TRMT10C-SDR5C1 complex during 3' processing by ELAC2, despite the slight inhibitory effect. We have extended the discussion of this phenomenon to clearly state the uncertainty regarding its mechanism and potential implications in vivo (lines 393 to 400).

In figure 1d, what is the band above pre-mt-tRNA^{Gln} (and to some extent pre-mt-tRNA^{Tyr})? Is there other cleavage sites/partial pre-cleavages occurring in these pre-mt-tRNA in vitro?

Unfortunately, we cannot determine for sure what this additional band corresponds to. As the reviewer correctly points out, it occurs in all pre-mt-tRNA^{Gln} preparations, as well as to some extent in most of the preparations of the other mt-tRNAs. As it migrates above the full-length pre-tRNA substrate, we do not think that it represents an alternative cleavage product. Also, as it is not cleaved by ELAC2 under any condition, we do not think it represents an in vitro transcription artifact due to non-templated 3'-extension, which should not interfere with 3'-trailer cleavage. Thus, we suspect that these bands represent impurities either due to incomplete digestion of DNA template (perhaps partially protected by hybridization to the RNA product), pre-tRNAs with extended 5'-end due to wrong transcription initiation or an entirely unrelated RNA product produced from contaminating DNA generated during the preparation of the template.

Cryo-EM structures of mt-RNase Z^{Tyr} and mt-RNase Z^{Gln} complexes (Figure 2): is one of the complexes more stable than the other? From cryo-EM processing & resolution reached, one structure appears more flexible than the other, but could the authors implement TSA or SPR assays to further explore whether the binding affinity of ELAC2 and/or TRMT10C for mt-tRNA^{Tyr} or mt-tRNA^{Gln} are different?

We thank the reviewer for this suggestion. In the revised manuscript, we include fluorescence anisotropy data that provide the relative affinities of ELAC2 and TRMT10C/SDR5C1 for the two mt-tRNA substrates. Interestingly, we find that TRMT10C/SDR5C1 binds both tRNAs with nearly the same affinity (with K_ds of 167 nM and 215 nM for mt-tRNA^{Gln} and mt-tRNA^{Tyr}, respectively). By contrast, ELAC2 has high affinity only for mt-tRNA^{Gln} (K_d of 301 nM), whereas its affinity for mt-tRNA^{Tyr} is significantly lower (K_d could not be determined accurately; appears to be >6 μM). The new data are included in the revised manuscript in Extended Data Figure 1c and lines 144-148.

Figure 3d, f and 5, d,f: local resolution of these parts of the maps are not entirely clear from the provided supplementary data. Please show the cryo-EM density around the atomic models to be more convincing (in supplementary data panels for instance)

We have included figure panels showing the cryo-EM density around atomic models and local resolutions, as suggested (Extended Data Figures 6e,f and 7e,f).

Molecular basis of human nuclear and mitochondrial tRNA 3'-processing

Bhatta, Kuhle, Yu et al.

Response to reviewer comments – Revision 2

Responses are in blue italics

Reviewer #1:

Remarks to the Author:

The revised version is acceptable for publication. I only have a minor comment. The methods should contain pH for each buffer. Some are missing, such as for the final cryo-EM sample before freezing the grids and for purifying the RNA.

We are grateful to the reviewer for dedicating their time and expertise to improving this manuscript and for providing insightful comments and constructive feedback on our work.

We have added pH for all buffers, including the final cryo-EM sample buffer and RNA purification buffer as suggested by the reviewer.

Reviewer #2:

Remarks to the Author:

The authors revised the manuscript, and the manuscript is now acceptable for publication in NSMB.

We thank the reviewer for their careful consideration of our manuscript, as well as for their thoughtful suggestions and positive evaluation of our research.

Reviewer #3:

Remarks to the Author:

Dr. Hillen and colleagues have integrated all feedback from the three reviewers into their revised manuscript. All of my concerns have been adequately addressed. I have no further comments to make, other than to note that that the work presented here is of a high standard and can be published in Nature Structural and Molecular Biology in its current form.

We thank the reviewer for their careful review and valuable inputs, and for appreciating the importance of our work through their encouraging comments and recommendations.